# Joint Representations for Reinforcement Learning with Multiple Sensors

## Abstract

Combining inputs from multiple sensor modalities effectively in reinforcement learning (RL) is an open problem. Many self-supervised representation learning approaches exist to improve performance and sample complexity for image-based RL. However, they typically do not consider other available information, such as robot proprioception, when learning the representation, but only concatenate it to independently learned image representations. Here, we show how using this proprioception for representation learning can help algorithms to focus on relevant aspects and guide them toward finding better representations. Building on *Recurrent State Space Models*, we systematically analyze representation learning approaches for RL from multiple sensors. We propose a novel combination of reconstruction-based and contrastive losses, which allows us to choose the most appropriate method for each sensor modality, and demonstrate its benefits in a wide range of settings. This evaluation includes model-free and model-based RL on complex tasks where the images contain distractions or occlusions, a new locomotion suite, and a visually realistic mobile manipulation task with both color and depth images. We show that learning a joint representation by combining contrastive and reconstruction-based losses significantly improves performance compared to the common practice of concatenating image representations with proprioception and allows solving more complex tasks that are beyond the reach of current SOTA representation learning methods.

## 1 Introduction

Learning compact representations of high-dimensional images has led to considerable advances in reinforcement learning (RL) from pixels. To date, most RL approaches that use representations (Hafner et al., 2019; 2020; Srinivas et al., 2020; Lee et al., 2020; Yarats et al., 2021b; Stooke et al., 2021; Zhang et al., 2020), learn them in isolation for a single high-dimensional sensor, such as a camera. However, while images are crucial to perceive an agent's surroundings in unstructured environments, they are often not the only available source of information. Most agents in realistic scenarios can also directly observe their internal state using sensors in their actuators, inertial measurement units, force and torque sensors, or other forms of proprioceptive sensing.

State Space Models (Murphy, 2012) naturally lend themselves to accumulating information across multiple sensors and time to form a single compact representation of the entire system state. By building on *Recurrent State Space Models (RSSMs)* (Hafner et al., 2019), this approach provides a scalable basis for RL in tasks with complex observations and dynamics. Previous work suggests using either reconstruction (Hafner et al., 2019; 2021) or contrastive methods (Hafner et al., 2020; Ma et al., 2020; Nguyen et al., 2021) to train *RSSMs*, both of which have their strengths and weaknesses. While reconstruction is a powerful tool as it forces models to capture the entire signal, it may fail to learn good representations if observations are noisy or contain distracting elements (Zhang et al., 2020; Ma et al., 2020; Deng et al., 2022). In such cases, contrastive methods can ignore irrelevant parts of the observation and still learn valuable representations. However, they are prone to representation collapse and struggle to learn the accurate dynamics required for model-based RL (Ma et al., 2020).

We propose combining contrastive and reconstruction-based approaches to leverage the benefits of both worlds. For example, reconstruction-based loss functions can be used for noiseless propri-

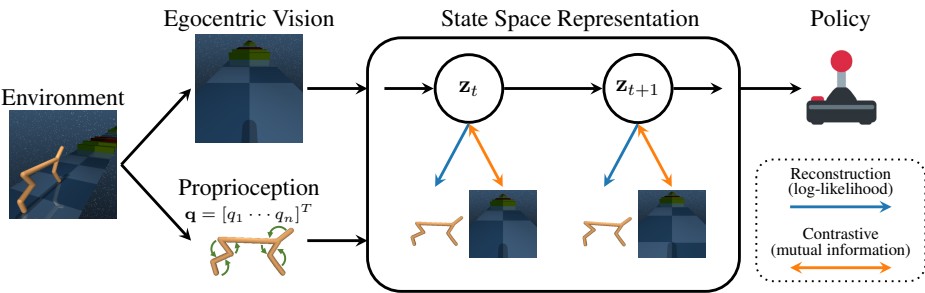

Figure 1: In this example, the agent has to jump over "hurdles" to move forward. It has to perceive the hurdles in its environment through egocentric vision but can directly observe its proprioceptive state, i.e., the position and velocity of its joints. We propose learning joint representations of all available sensors using a combination of reconstruction-based and contrastive objectives. This approach allows us to use reconstruction for clean low-dimensional sensors, e.g., proprioception, and contrastive losses for high-dimensional noisy sensor signals such as images. We build on *Recurrent State Space Models* to accumulate information across sensors and time and use our representations for model-free and model-based RL.

oception and a contrastive loss for images with distractors, where reconstruction fails (Ma et al., 2020; Nguyen et al., 2021). Fig. 1 shows an overview of this approach. The common approach to training *RSSMs* is variational inference (VI), which provides a basis for both reconstruction and contrastive objectives. In the original formulation (Hafner et al., 2019), *RSSMs* are trained with VI using pure reconstruction. However, the reconstruction terms can be replaced with contrastive losses based on mutual information estimation (Hafner et al., 2020; Ma et al., 2020). Contrastive predictive coding (CPC) (Oord et al., 2018) offers an alternative to the variational approach of training *RSSMs* (Nguyen et al., 2021; Srivastava et al., 2021). These methods train the *RSSMs'* system dynamics by maximizing the agreement of predicted future latent states with future observations. Since the recent literature is inconclusive about whether the variational or the predictive approach is preferable, we evaluate our representation learning using both paradigms.

We build our representation learning method into model-free and model-based RL agents and systematically evaluate the effects of learning a joint representation on tasks from the DeepMind Control (DMC) Suite (Tassa et al., 2018). To further investigate the approaches' capabilities, we use modified DMC tasks with *Video Background* (Zhang et al., 2020; Nguyen et al., 2021) and *Occlusions*. Additionally, we evaluate on a new *Locomotion* suite and a visually realistic mobile manipulation task, where agents must combine proprioception and egocentric vision to move, navigate, and interact with the environment. For the mobile manipulation tasks, we consider both color and depth images to demonstrate that our methods apply to different visual modalities. Our experiments show that joint representations improve performance over learning an image-only representation and concatenating it with proprioception. In the task with *Video Background*, using a combination of contrastive and reconstruction losses enables us almost to match the performance of current SOTA methods on standard images. The *Occlusion* task is out of reach for image-only approaches and can only be solved by appropriately combining images and proprioception. Moreover, we show that joint representations improve the performance of model-based agents with contrastive image representations, which are known to perform worse than reconstruction-based approaches (Hafner et al., 2020; Ma et al., 2020).

To summarize our contributions, we propose a general framework for training joint representations based on *RSSMs* by combining contrastive and reconstruction losses based on the properties of the individual sensor. This framework contains objectives motivated by a variational and a contrastive predictive coding viewpoint. We conduct a large-scale evaluation using model-free and model-based approaches and show that using joint representations significantly increases performance over concatenating image representations and proprioception. Further, they help to learn better models for model-based RL when a contrastive image loss is required. We introduce DMC tasks with *Occlusions* and a *Locomotion* suite as new challenges for representation learning in RL. On these tasks, as well as DMC tasks with *Video Background* and a mobile manipulation task our approach outperforms several SOTA baselines and allows solving tasks where image-only approaches fail.

## 2 RELATED WORK

**Representations for Reinforcement Learning.** Many recent approaches use ideas from genera- tive (Wahlström et al., 2015; Watter et al., 2015; Banijamali et al., 2018; Lee et al., 2020; Yarats et al., 2021b) and self-supervised representation learning (Zhang et al., 2020; Srinivas et al., 2020; Yarats et al., 2021a; Stooke et al., 2021; You et al., 2022) to improve performance, sample efficiency, and generalization of RL from images. Those based on *Recurrent State Space Models (RSSMs)* are particularly relevant for this work. When proposing the *RSSM*, (Hafner et al., 2019) used a genera- tive approach. They formulated their objective as auto-encoding variational inference, which trains the representation by reconstructing observations. Such reconstruction-based approaches have lim- itations with observations containing noise or many task-irrelevant details. As a remedy, Hafner et al. (2020) proposed a contrastive alternative based on mutual information and the InfoNCE esti- mator (Poole et al., 2019). Ma et al. (2020) refined this approach and improved results by modifying the policy learning mechanism. Using a different motivation, namely contrastive predictive cod- ing (Oord et al., 2018), Okada & Taniguchi (2021); Nguyen et al. (2021); Srivastava et al. (2021); Okada & Taniguchi (2022) proposed alternative contrastive learning objectives for *RSSMs*. In this work, we leverage the variational and predictive coding paradigms and show that joint representa- tions improve performance for both. Fu et al. (2021); Wang et al. (2022) propose further factorizing the *RSSM*'s latent variable to disentangle task-relevant and task-irrelevant information. However, unlike contrastive approaches, they explicitly model the task-irrelevant parts instead of ignoring them, which can impede performance if the distracting elements become too complex to model. Other recent approaches for learning *RSSMs* include using prototypical representations (Deng et al., 2022) or masked reconstruction (Seo et al., 2022). Out of these works, only Srivastava et al. (2021) consider using additional proprioceptive information. Yet, they did so only in a single experiment and did not investigate a combination of reconstruction and contrastive losses.

**Sensor Fusion in Reinforcement Learning.** Many application-driven approaches to visual RL for robots use proprioception to solve their specific tasks (Finn et al., 2016; Levine et al., 2016; Kalashnikov et al., 2018; Xiao et al., 2022; Fu et al., 2022). Yet, they usually do not use dedicated representation learning or concatenate image representations and proprioception. Several notable exceptions use *RSSMs* with images and proprioception and thus learn joint representations (Wu et al., 2022; Hafner et al., 2022; Becker & Neumann, 2022; Hafner et al., 2023). Seo et al. (2023) learn world models using multiple images from different viewpoints. However, all of them focus on a purely model-based setting and do not investigate joint-representation learning with *RSSMs* as an alternative to concatenation for model-free RL. Additionally, they only consider reconstruction- based objectives, while we emphasize contrastive and especially combined methods.

**Multimodal Representation Learning.** Representation learning from multiple modalities has widespread applications in general machine learning, where methods such as *CLIP* (Radford et al., 2021) combine language concepts with the semantic knowledge of images and allow language-based image generation (Ramesh et al., 2022). For robotics, Brohan et al. (2022); Mees et al. (2022); Driess et al. (2023); Shridhar et al. (2022; 2023) combine language models with the robot's perception for natural language-guided manipulation tasks using imitation learning. In contrast, we work in an online RL setting and mainly consider different modalities, namely images and proprioception.

## 3 STATE SPACE MODELS FOR JOINT REPRESENTATION LEARNING

Given trajectories of observations $\mathbf{o}_{1:T} = \{\mathbf{o}_t\}_{t=1:T}$ and actions $\mathbf{a}_{1:T} = \{\mathbf{a}_t\}_{t=1:T}$ we aim to learn a state representation that is well suited for RL. We assume the observations stem from $K$ different sensors, $\mathbf{o}_t = \{\mathbf{o}_t^{(k)}\}_{k=1:K}$, where the individual $\mathbf{o}_t^{(k)}$ only contain partial information about the system state. Further, even $\mathbf{o}_t$ may not contain all necessary information for optimal acting, i. e., the environment is partially observable, and the representation has to accumulate information over time.

Our goal is to learn a concise, low dimensional representation $\phi(\mathbf{o}_{1:t}, \mathbf{a}_{1:t-1})$ that accumu- lates all relevant information until time step $t$. We provide this representation to a policy $\pi(\mathbf{a}_t | \phi(\mathbf{o}_{1:t}, \mathbf{a}_{1:t-1}))$ which aims to maximize the expected return in a given RL problem. Here, we have a cyclic dependency, as the policy collects the trajectories to learn the representation by acting in the environment. In this setting, the policy's final return and the sample complexity of the entire system determine what constitutes a *good* representation.

State Space Models (SSMs) (Murphy, 2012) naturally lend themselves to sensor fusion and information accumulation problems. We assume a latent state variable, $\mathbf{z}_t$, which evolves according to a Markovian dynamics $p(\mathbf{z}_{t+1}|\mathbf{z}_t, \mathbf{a}_t)$ given an action $\mathbf{a}_t$. At each time step $t$, each of the $K$ observations is generated from the latent state by an observation model $p^{(k)}(\mathbf{o}_t^{(k)}|\mathbf{z}_t)$. The initial state is distributed according to $p(\mathbf{z}_0)$. Here, the belief over the latent state, taking into account all previous actions as well as previous and current observations $p(\mathbf{z}_t|\mathbf{a}_{1:t-1}, \mathbf{o}_{1:t})$ can be used as the representation. Yet, computing $p(\mathbf{z}_t|\mathbf{a}_{1:t-1}, \mathbf{o}_{1:t})$ analytically is intractable for models of relevant complexity and we use variational approximation $\phi(\mathbf{o}_{1:t}, \mathbf{a}_{1:t-1}) \hat{=} q(\mathbf{z}_t|\mathbf{a}_{1:t-1}, \mathbf{o}_{1:t})$. This variational approximation also plays an integral part during training and is thus readily available as input for the policy.

We instantiate the generative SSM and the variational distribution using a *Recurrent State Space Model (RSSM)* (Hafner et al., 2019), which splits the latent state $\mathbf{z}_t$ into a stochastic and a deterministic part. Following (Hafner et al., 2019; 2020), we assume the stochastic part of the *RSSM*'s latent state to be Gaussian. While the original *RSSM* only has a single observation model $p(\mathbf{o}_t|\mathbf{z}_t)$, we extend it to $K$ models, one for each observation modality. The variational distribution takes the deterministic part of the state together with the $K$ observations $\mathbf{o}_t = \{\mathbf{o}_t^{(k)}\}_{k=1:K}$ and factorizes as $q(\mathbf{z}_{1:t}|\mathbf{o}_{1:t}, \mathbf{a}_{1:t-1}) = \prod_{t=1}^{T} q(\mathbf{z}_t|\mathbf{z}_{t-1}, \mathbf{a}_{t-1}, \mathbf{o}_t)$. To account for multiple observations instead of one, we first encode each observation individually using a set of $K$ encoders, concatenate their outputs, and provide the result to the *RSSM*. Finally, we also learn a reward model $p(r_t|\mathbf{z}_t)$ to predict the reward from the representation. Following the findings of Srivastava et al. (2021) and Tomar et al. (2023) we also include reward prediction to learn the representations for model-free agents.

## 3.1 Learning the State Space Representation

We combine reconstruction-based and contrastive approaches to train our representations. Training *RSSMs* can be based on either a variational viewpoint (Hafner et al., 2020; Ma et al., 2020) or a contrastive predictive coding (Oord et al., 2018) viewpoint (Nguyen et al., 2021; Srivastava et al., 2021). We investigate both approaches, as neither decisively outperforms the other.

**Variational Learning.** Originally, (Hafner et al., 2019) proposed leveraging a fully generative approach for *RSSMs*. Building on the stochastic variational autoencoding Bayes framework (Kingma & Welling, 2013; Sohn et al., 2015), they derive a variational lower bound objective. After inserting our assumption that each observation factorizes into $K$ independent observations and adding a term for reward prediction, this objective is given as

$$\sum_{t=1}^{T} \mathbb{E}\left[ \sum_{k=1}^{K} \log p^{(k)}(\mathbf{o}_t^{(k)}|\mathbf{z}_t) + \log p(r_t|\mathbf{z}_t) - \mathrm{KL}\left[ q(\mathbf{z}_t|\mathbf{z}_{t-1}, \mathbf{a}_{t-1}, \mathbf{o}_t) \parallel p(\mathbf{z}_t|\mathbf{z}_{t-1}, \mathbf{a}_{t-1}) \right] \right], \quad (1)$$

where the expectation is formed over the distribution $p(\mathbf{o}_{1:t}, \mathbf{a}_{1:t-1}) q(\mathbf{z}_t|\mathbf{o}_{1:t}, \mathbf{a}_{1:t-1})$, i.e., sub-trajectories from a replay buffer and the variational distribution. Optimizing this bound using the reparametrization trick (Kingma & Welling, 2013; Rezende et al., 2014) and stochastic gradient descent simultaneously trains the variational distribution and all parts of the generative model. While this approach can be highly effective, reconstructing high-dimensional, noisy observations can also cause issues. First, it requires introducing large observation models. These observation models are unessential for the downstream task and are usually discarded after training. Second, the reconstruction forces the model to capture all details of the observations, which can lead to highly suboptimal representations if images are noisy or contain task-irrelevant distractions.

**Contrastive Variational Learning** (CV) can provide a remedy to these problems. To introduce such contrastive terms, we can replace the individual log-likelihood terms with mutual information (MI) terms $I(\mathbf{o}_t^{(k)}, \mathbf{z}_t)$ by adding and subtracting $\log p(\mathbf{o}^{(k)})$ (Hafner et al., 2020; Ma et al., 2020)

$$\mathbb{E}\left[ \log p^{(k)}(\mathbf{o}_t^{(k)}|\mathbf{z}_t) \right] = \mathbb{E}\left[ \log \frac{p^{(k)}(\mathbf{o}_t^{(k)}|\mathbf{z}_t)}{p(\mathbf{o}_t^{(k)})} + \log p(\mathbf{o}_t^{(k)}) \right] = \mathbb{E}\left[ I(\mathbf{o}_t^{(k)}, \mathbf{z}_t) \right] + c. \quad (2)$$

Intuitively, the MI measures how informative a given latent state is about the corresponding observations. Thus, maximizing it leads to similar latent states for similar sequences of observations and actions. While we cannot analytically compute the MI, we can estimate it using the InfoNCE bound (Oord et al., 2018; Poole et al., 2019). Doing so eliminates the need for generative reconstruction and instead only requires a discriminative approach based on a score function

$f_v^{(k)}(\mathbf{o}_t^{(k)}, \mathbf{z}_t) \mapsto \mathbb{R}_+$. This score function measures the compatibility of pairs of observations and latent states. It shares large parts of its parameters with the *RSSM*. For details on the exact parameterization, we refer to Appendix B. This methodology allows the mixing of reconstruction and mutual information terms for the individual sensors, resulting in a generalization of Equation 1,

$$\sum_{t=1}^{T}\sum_{k=1}^{K} \mathcal{L}_v^{(k)}(\mathbf{o}_t^{(k)}, \mathbf{z}_t) + \mathbb{E}\left[\log p(r_t|\mathbf{z}_t) - \mathrm{KL}\left[q(\mathbf{z}_t|\mathbf{z}_{t-1}, \mathbf{a}_{t-1}, \mathbf{o}_t) \parallel p(\mathbf{z}_t|\mathbf{z}_{t-1}, \mathbf{a}_{t-1})\right]\right]. \quad (3)$$

Here $\mathcal{L}_v^{(k)}$ is either the log-likelihood or the MI term. As we show in Section 4 choosing the terms corresponding to the properties of the corresponding modality can often improve performance.

**Contrastive Predictive Coding** (CPC) (Oord et al., 2018) provides an alternative to the variational approach. The idea is to maximize the MI between the latent variable $\mathbf{z}_t$ and the next observation $\mathbf{o}_{t+1}^{(k)}$, i.e., $I(\mathbf{o}_{t+1}^{(k)}, \mathbf{z}_t)$. While this approach seems similar to contrastive variational learning, there is a crucial difference. We estimate the MI between the current latent state and the next observation, not the current observation. Thus, we explicitly predict ahead to compute the loss. As we use the *RSSM's* dynamics model for the prediction, this formalism provides a training signal to the dynamics model. However, Levine et al. (2019); Shu et al. (2020); Nguyen et al. (2021) discuss how this signal alone is insufficient for model-based RL. Srivastava et al. (2021) show that similar ideas also benefit model-free RL and we follow their approach by regularizing the objective using KL-term from Equation 1 weighted with a small factor $\beta$. Additionally, we can turn individual contrastive MI terms into reconstruction terms for suitable sensor modalities by reversing the principle of Equation 2. Including reward prediction, this results in the following maximization objective

$$\sum_{t=1}^{T}\sum_{k=1}^{K} \mathcal{L}_p^{(k)}(\mathbf{o}_{t+1}^{(k)}, \mathbf{z}_t) + \mathbb{E}\left[\log p(r_t|\mathbf{z}_t) - \beta\mathrm{KL}\left[q(\mathbf{z}_t|\mathbf{z}_{t-1}, \mathbf{a}_{t-1}, \mathbf{o}_t) \parallel p(\mathbf{z}_t|\mathbf{z}_{t-1}, \mathbf{a}_{t-1})\right]\right], \quad (4)$$

where $\mathcal{L}_p^{(k)}$ is either the one-step ahead likelihood $\log p(\mathbf{o}_{t+1}^{(k)}|\mathbf{z}_t)$ or an InfoNCE estimate of $I(\mathbf{o}_{t+1}^{(k)}, \mathbf{z}_t)$ using a score function $f_p^{(k)}(\mathbf{o}_{t+1}^{(k)}, \mathbf{z}_t) \mapsto \mathbb{R}_+$. From an implementation viewpoint, the resulting approach differs only slightly from the variational contrastive one. For CPC approaches, we use a sample from the *RSSM's* dynamics $p(\mathbf{z}_{t+1}|\mathbf{z}_t, \mathbf{a}_t)$ and for contrastive variational approaches we use a sample from the variational distribution $q(\mathbf{z}_t|\mathbf{z}_{t-1}, \mathbf{a}_{t-1}, \mathbf{o}_t)$.

**Estimating Mutual Information with InfoNCE.** We estimate the mutual information (MI) using $b$ mini-batches of sub-sequences of length $l$. After computing the latent estimates, we get $I = b \cdot l$ pairs $(\mathbf{o}_i, \mathbf{z}_i)$, i.e., we use both samples from the elements of the batch as well as all the other time steps within the sequence as negative samples. Using those, the symmetry of MI, the InfoNCE bound (Poole et al., 2019), and either $f = f_v^{(k)}$ or $f = f_p^{(k)}$, we can estimate the MI as

$$0.5\left(\sum_{i=1}^{I} \log \frac{f(\mathbf{o}_i, \mathbf{z}_i)}{\sum_{j=1}^{I} f(\mathbf{o}_i, \mathbf{z}_j)} + \log \frac{f(\mathbf{o}_i, \mathbf{z}_i)}{\sum_{j=1}^{I} f(\mathbf{o}_j, \mathbf{z}_i)}\right).$$

### 3.2 Learning to Act Based on the Representation

We consider both model-free and model-based reinforcement learning. For the former, we use Soft Actor-Critic (SAC) (Haarnoja et al., 2018) on top of the representation by providing the deterministic part of the latent state and the mean of the stochastic part as input to both the actor and the critic. For the latter, we use *latent imagination* (Hafner et al., 2020), which propagates gradients through the learned dynamics model to optimize the actor. In both cases, we alternatingly update the *RSSM*, actor, and critic for several steps before collecting a new sequence in the environment. The *RSSM* uses only the representation learning loss and gets neither gradients from the actor nor the critic.

## 4 Experiments

We evaluate our representation learning approaches on 4 task suites and a mobile manipulation task, introduced below. For each suite, we present the final aggregated performance using Interquartile Means (IQM) and 95% Stratified Bootstrapped Confidence Intervals (CIs) (Agarwal et al., 2021).

Appendix A provides details about all considered tasks. Appendix B lists all hyperparameters. Appendix C shows learning curves for all representation learning paradigms on all tasks, performance profiles, and per-environment results. Code is available[1]. Following prior work (Srivastava et al., 2021; Deng et al., 2022), we include cropping-based image augmentation for contrastive approaches.

**Representation Learning Methods.** We name our approaches for joint representation learning by combining a contrastive image loss with reconstruction for the proprioception *Joint(CV+R)* and *Joint(CPC+R)*. Those build on the contrastive variational (Equation 3) and the contrastive predictive coding (Equation 4) objectives respectively. Additionally, we consider *Joint(R+R)*, a purely reconstructive approach based on Equation 1.

We include several SOTA visual RL approaches to show the competitiveness of our approach and how including proprioception allows us to solve problems that challenge more tailored approaches. First, we consider the model-free *DrQ-v2* (Yarats et al., 2022) and extend it to also use proprioception (*DrQ-v2(I+P)*). Both learn no explicit representations. Further, we include the reconstruction-based *Dreamer-v3* (Hafner et al., 2023) (model-based), the self-supervised *DreamerPro* (Deng et al., 2022) (model-based), and *DenoisedMDP* (Wang et al., 2022) (model-free and model-based). We also consider *DBC* (Zhang et al., 2020)(model-free) and *TIA* (Fu et al., 2021)(model-based). Except *DrQ-v2(I+P)* these baselines work solely on images.

To show the benefits of combining contrastive and reconstruction-based objectives, we compare with variants that use contrastive losses for both modalities, named *Joint(CV+CV)* and *Joint(CPC+CPC)*. We also compare to the naive approach of concatenating proprioception to image representations and use either reconstruction (*Concat(R)*) or a contrastive method (*Concat(CV)* or *Concat(CPC)*) to train the image representation. These are not applicable in the model-based setting, as they cannot predict future proprioception. We also include single-sensor approaches. For images, we again use all three representation learning approaches, resulting in *Img-Only(R)*, *Img-Only(CV)*, and *Img-Only(CPC)*. For the proprioception-only agents, we use SAC (Haarnoja et al., 2018) directly on the proprioception (*ProprioSAC*). From these approaches, model-based *Img-Only(R)* corresponds largely to *Dreamer-v1* (Hafner et al., 2020) and model-free *Img-Only(CPC)* and *Joint(CPC+CPC)* resemble the approach introduced in (Srivastava et al., 2021). See Appendix B.5 for details.

**Tasks.** We use seven tasks from the DeepMind Control Suite (DMC) (Tassa et al., 2018) that cover a wide range of challenges, namely `Ball-in-Cup Catch`, `Cartpole Swingup`, `Cheetah Run`, `Reacher Easy`, `Walker Walk`, `Walker Run`, and `Quadruped Walk`. We split their states into proprioceptive entries and non-proprioceptive entries such that the proprioception only contains partial information of the full environment state. The remaining information has to be inferred from images. For example, in `Ball-in-Cup Catch` the cup's state is proprioceptive while the ball's state is not. Besides using the *Standard Images*, we add *Video Backgrounds* or *Occlusions for all seven tasks, creating a total of 3 different task suites*. For *Video Backgrounds*, we follow (Nguyen et al., 2021; Deng et al., 2022) and render videos from the Kinetics400 dataset (Kay et al., 2017) behind the agent. For *Occlusions*, we add slowly moving disks in front of the agent. The upper row of Fig. 6 shows examples. In these tasks, the challenge lies in learning representations that filter out irrelevant visual details while focusing on relevant aspects. Further, *Occlusions* tests the approaches' capabilities to maintain a consistent representation across time under partial observability, which increases the task's difficulty considerably.

In addition, we propose a novel *Locomotion* suite consisting of six tasks. All tasks include obstacles that have to be localized through egocentric vision in order to be avoided. As the agents cannot observe themselves from the egocentric perspective, they additionally need proprioception. sFig. 1 shows examples for the modified `Cheetah Run`. These tasks test the representations' ability to combine information from both sources to enable successful navigation and movement.

Further, we build on the *OpenCabinetDrawer* task from ManiSkill2 (Gu et al., 2023) where a mobile robot navigates to a cabinet and opens a drawer using egocentric vision and proprioception. The robot and a randomly sampled drawer are placed in a realistic apartment setting. We evaluate two separate scenarios, one with constant and one with changing lighting and surroundings. Additionally, we evaluate two visual modalities, i.e., standard RGB images and Depth images. Both types of images are from the same egocentric perspective. The task's complexity stems from the large action space and visual realism. Fig. 5 provides examples of the changing conditions setting.

---

[1] *<supplement, GitHub link will be added here>*

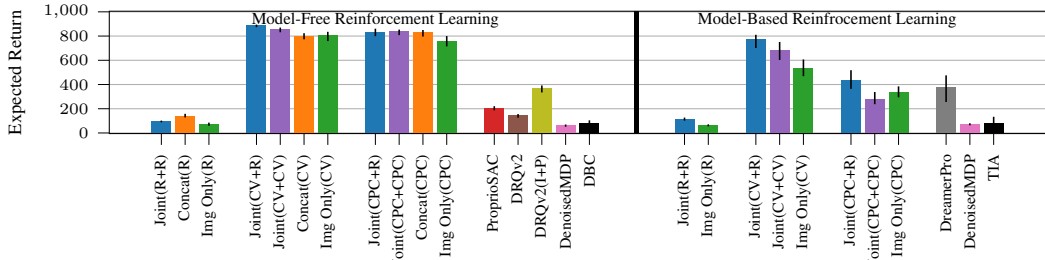

Figure 2: Aggregated performance after $10^6$ environment steps on the *Video Background* suite (IQM and 95% CIs). Even with *Video Backgrounds*, model-free *Joint(CV+R)* reaches the performance of SOTA approaches on *Standard Images* (Fig. 8, Fig. 9) and outperforms all model-free and model-based baselines. Model-based *Joint(CV+R)* improves performance over its fully contrastive counterpart and DreamerPro. It almost matches the model-free performance. While CPC approaches perform similarly to CV approaches for model-free RL, they are worse for model-based RL.

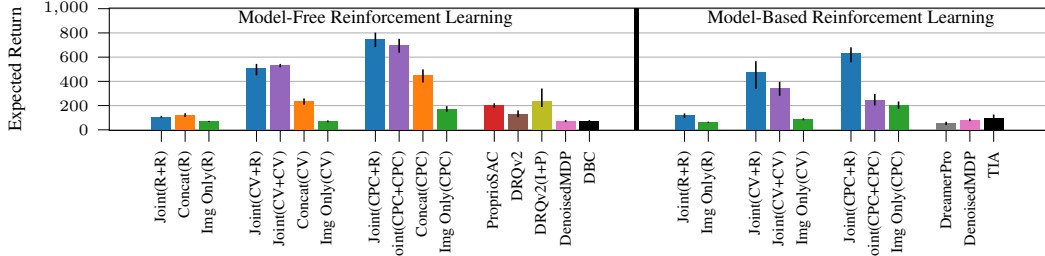

Figure 3: Aggregated performance after $10^6$ environment steps on the *Occlusion* suite (IQM and 95% CIs). Model-free *Joint(CPC+R)* performs best among all considered approaches, significantly outperforming all concatenation-based approaches. For model-based *Joint(CPC+R)* more than doubles the score than its fully contrastive counterpart, showing how using reconstruction for low dimensional sensors can remedy issues with contrastive learning in model-based RL. No approach using image reconstruction or only a single modality achieves reasonable performance.

**Discussion.** Our analysis builds on the full results presented in Appendix C. Our experiments show how joint representations outperform the concatenation of image representations with proprioception, single-senor baselines, and a large selection of model-free and model-based baselines. These effects are pronounced for the more difficult settings, i.e., *Occlusions* (Fig. 3), *Locomotion* (Fig. 4), and *OpenCabinetDrawer* (Fig. 5) Furthermore, our combination of contrastive and reconstruction-based losses (*Joint(CV+R)* and *Joint(CPC+R)*) outperforms using purely contrastive objectives (*Joint(CV+CV)* and *Joint(CPC+CPC)*), in particular for model-based RL and the harder tasks.. In both *Occlusions* (Fig. 3) and *OpenCabinetDrawer* (Fig. 5), no image-only approach learns reasonable behavior or manages to outperform *ProprioSAC*. While the *Concat* approaches already improve performance using joint representation gives further significant gains in both sample complexity (Fig. 8, Fig. 9, Fig. 13, Fig. 14) and final performance. Our experiments on *OpenCabinetDrawer* (Fig. 5), show these effects also hold when using depth images instead of color images.

While model-free and model-based agents perform similarly well for approaches that reconstruct images, model-based agents perform worse than their model-free counterparts for contrastive image losses (Fig. 2, Fig. 3, Fig. 8, Fig. 9). This is in line with previous findings (Ma et al., 2020), indicating that contrastive approaches struggle to learn suitable long-term dynamics that enable successful model-based RL. Notably, for the image-only representations, this performance gap is significantly larger than for joint representations. Using a joint representation almost closes the gap between model-free and model-based for *Joint(CV+R)* and allows the *Joint(CPC+R)* to significantly outperform the proprioception-only baseline (Fig. 2, Fig. 3). This result demonstrates how joint representations allow learning of stable long-term dynamics that enable more successful model-based RL. However, we still find that model-free methods perform better for contrastive representations and thus only consider those for the *Locomotion* suite and *OpenCabinetDrawer*.

When the images contain no irrelevant aspects, such as the *Standard Images*, reconstruction-based approaches perform on par with their contrastive counterparts in the model-free setting (Fig. 8) and outperform them in the model-based setting (Fig. 9). Yet, as expected, approaches that reconstruct

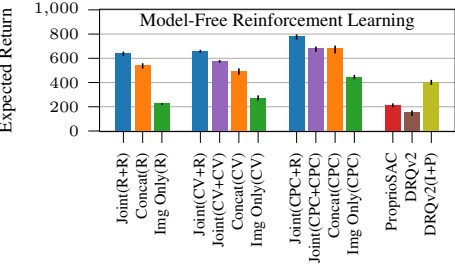

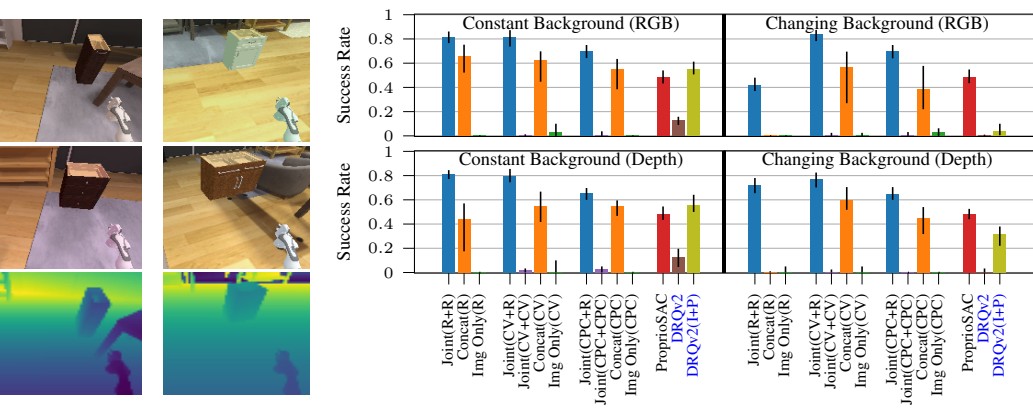

Figure 4: Aggregated performance after $10^6$ environment steps on the *Locomotion* suite (IQM and 95% CIs). *Joint(CPC+R)* significantly outperforms all concatenation-based approaches, highlighting how joint representation approaches can extract and combine information more effectively. It also outperforms image reconstruction(*Joint(R+R)*), which is noteworthy as the *Locomotion* suite tasks do not explicitly contain distracting elements.

Figure 5: **Left**: Example color and depth images from *OpenCabinetDrawer* with changing background. **Right**: Performance of model-free agents after $1.5 \times 10^6$ environment steps on *OpenCabinetDrawer* (IQM and 95% CIs) with constant and changing background. These results further show the benefits of learning joint representations with appropriate losses for each modality. The fully contrastive approaches ( *Joint(CV+CV)* and *Joint(CPC+CPC)* fail on both tasks and, for color images *Joint(R+R)* only works with constant background. *Joint(CV+R)* and *Joint(CPC+R)* on the other hand perform well in both settings, outperforming concatenation-based approaches and maintaining their performance when switching to the more challenging changing background setting. When using depth images, we get generally similar results. Yet, the performance decreases less for *Joint(R+R)*, the contrastive concatenation baselines, and *DrQv2(I+P)* when adding the changing backgrounds as using depth images removes some sources of noise, such as the changing lighting.

images fail in the *Video Background* (Fig. 2) and *Occlusion* (Fig. 3) tasks as well as the changing background variant of *OpenCabinetDrawer* (Fig. 5). Here, only joint representations with a contrastive image loss and reconstruction for proprioception perform well, as they can ignore the irrelevant aspects of the images. In the *Locomotion* experiments, the CPC approaches (Fig. 4) have a significant edge over reconstruction. While highly relevant to the task, the obstacles still appear at random and have random colors for some tasks, which makes reconstruction harder. The contrastive methods' advantage is pronounced in those tasks with random colored obstacles. When comparing the contrastive learning paradigms, the contrastive variational approach has an edge over CPC on *Video Backgrounds* (Fig. 2) and *OpenCabinetDrawer* (Fig. 5) with color and depth images. Yet, CPC performs better in *Occlusion* (Fig. 3) and *Locomotion* (Fig. 4). In particular, the better performance in *Occlusions* indicates CPC approaches are better suited for propagating information over multiple time steps. However, our method of learning a joint representation using a contrastive image loss and reconstruction for proprioception performs best in both paradigms across all tasks.

Using depth instead of color for *OpenCabinetDrawer* (Fig. 5) leads to similar conclusions with constant backgrounds. With the more challenging changing backgrounds, using depth images removes some noise sources, e.g., the changing lighting. This allows several approaches to achieve higher performance but has only minor effects on the ranking, with *Joint(CV+R)* performing best for both image types, which shows how joint representations are beneficial across different visual modalities.

When comparing to SOTA image-only methods, model-based *Img-Only(R)*, which strongly resembles *Dreamer-v1*, cannot quite match the performance of *Dreamer-v3* and *DreamerPro* on *Standard Images* (Fig. 9). Yet, the contrastive *Img-Only* approaches outperform all other image-only baselines on *Video Backgrounds* (Fig. 2) and *Occlusions* (Fig. 3). *DreamerPro* retains some performance on

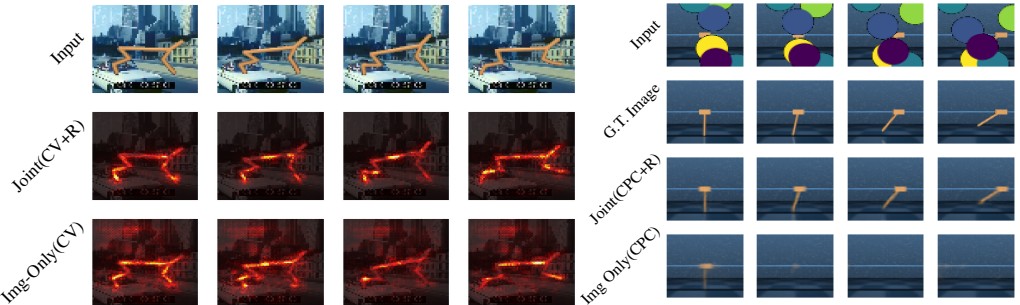

Figure 6: **Left:** Saliency Maps showing on which pixels the respective representation learning approaches focus. *Joint(CV+R)* clearly focuses better on the task-relevant cheetah, while *Img-Only(CV)* is more distracted by the video background. **Right:** In the occlusion tasks, we train a separate decoder to reconstruct the occlusion-free ground truth from the (detached) latent representation. For `Cartpole Swingup` only the cart position is part of the proprioception. Still, *Joint(CPC+R)* can capture both cart position and pole angle, while *Img-Only(CPC)* fails to do so.

*Video Backgrounds*, yet, like all other image-only approaches, it fails on *Occlusions*. Similar to the results of (Stone et al., 2021), we find that *DrQ-v2*'s performance deteriorates when adding *Video Backgrounds* or the changing backgrounds for *OpenCabinetDrawer*. Including proprioception helps for *DrQ-v2*, in particular in *Locomotion* (Fig. 4) and *OpenCabinetDrawer* (Fig. 5), yet it still performs worse than that of our approach and even the *Concat* baselines. *DBC*, *TIA*, and *DenoisedMDP* fail in both *Video Backgrounds* and *Occlusions*. The discrepancy in performance to similar video background tasks is due to a more difficult experimental setup, detailed in Appendix B.5. These results clearly show that including readily available proprioception allows solving visually more challenging tasks and that doing so naively (*DRQ-v2(I+P)* and *Concat*) is suboptimal.

The performance profiles (Fig. 10, Fig. 11, Fig. 12) show that the aggregated performance underlying our analysis is representative of the per-task performance, i.e., if an approach outperforms another when considering the aggregated performance, it also does so on a large majority of the individual tasks. Furthermore, performance is consistent across the different observation types for the *DMC* tasks, i.e., *Occlusions* are more difficult than *Video Background*, which are more difficult than *Standard Images*. Finally, we qualitatively investigate some of the learned representations in Fig. 6, which illustrates how joint representation learning can help the approaches to focus on relevant aspects and extract all necessary information from an image.

## 5  CONCLUSION

We consider the problem of Reinforcement Learning (RL) from multiple sensors, in particular images and proprioception. Building on *Recurrent State Space Models*, we learn joint representations of all available sensors instead of considering them in isolation. We propose combining contrastive and reconstruction approaches and consider variational and predictive coding paradigms for training. Our large-scale evaluation on modified versions of the DeepMind Control Suite, a novel *Locomotion* suite, and a visually realistic mobile manipulation task with both color and depth images, shows the benefits of this approach. We distill the results of this evaluation into the following takeaways: (**i**) Joint representations outperform the concatenation of image representations and proprioception. (**ii**) Combining contrastive approaches for images with reconstruction for low-dimensional, concise signals can significantly improve performance, especially in harder tasks. Yet, both the contrastive variational and contrastive predictive paradigms deserve consideration, as they perform differently well in different settings. (**iii**) In model-based RL, joint representations offer an easy and highly effective way to improve performance in tasks that require contrastive image objectives.

**Limitations.** While we showed the benefits of joint representations in both cases, our evaluation is inconclusive about whether contrastive variational or contrastive predictive coding approaches are generally preferable. Here, further investigation is required to deepen the understanding of their advantages and disadvantages. Additionally, even with a combination of contrastive and reconstruction losses, model-free agents perform better than their model-based counterparts. This suggests there is still room for improvement in contrastive *RSSM* training, especially w.r.t. dynamics learning.

**Reproducibility Statement.** We follow the suggestions from Agarwal et al. (2021) to provide information about the statistical significance of our results. Appendix A provides details about all newly created and modified environments and tasks. Appendix B lists all hyperparameters and architectural details of our model and further information on the used baselines. Appendix C lists the full results underlying the analysis in this paper and further information on the result aggregation and numbers of seeds. Code to reproduce all our results is available in the supplement and will be published after de-anonymization.

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

Table 1: Splits of the entire system state into proprioceptive and non-proprioceptive parts for the DeepMind Control Suite environments.

| Environment | Proprioceptive | Non-Proprioceptive |
|---|---|---|
| Ball In Cup | cup position and velocity | ball position and velocity |
| Cartpole | cart position and velocity | pole angle and velocity |
| Cheetah | joint positions and velocities | global pose and velocity |
| Reacher | reacher position and velocity | distance to target |
| Quadruped | joint positions and velocities | global pose + velocity, forces |
| Walker | orientations and velocities of links | global pose and velocity, height above ground |

Table 2: Splits of the entire system state into proprioceptive and non-proprioceptive parts for the Locomotion Suite. Some of the agents (Cheetah, Walker, Quadruped) require more proprioceptive information for the locomotion tasks with an egocentric vision than for the standard tasks with images from an external perspective.

| Environment | Proprioceptive | Non-Proprioceptive |
|---|---|---|
| Ant | joint position and velocity global velocities | wall positions global position |
| Hurdle Cheetah | joint positions and velocities global velocity | hurdle positions and heights global position |
| Hurdle Walker | orientations and velocities of links | hurdle positions and height global position and velocity |
| Quadruped (Escape) | joint positions and velocities, torso orientation and velocity, imu, forces, and torques at joints | Information about terrain |

## A  ENVIRONMENTS

### A.1  DEEPMIND CONTROL SUITE TASKS

Table 1 states how we split the states of the original DeepMind Control Suite (DMC) (Tassa et al., 2018) tasks into proprioceptive and non-proprioceptive parts. For the model-based agents, we followed common practice (Hafner et al., 2020; Fu et al., 2021; Wang et al., 2022; Deng et al., 2022) and use an action repeat of 2 for all environments. We do the same for the model-free agents except for: `Ball In Cup Catch` (4), `Cartpole Swingup` (8), `Cheetah Run` (4) and `Reacher Easy` (4). All environments in the locomotion suite also use an action repeat of 2, this includes `Hurdle Cheetah Run` which requires more fine-grained control than the normal version to avoid the hurdles.

**Natural Background.**  Following (Nguyen et al., 2021; Deng et al., 2022; Zhang et al., 2020; Fu et al., 2021; Wang et al., 2022) we render videos from the `driving car` class of the Kinetics400 dataset (Kay et al., 2017) behind the agents to add a natural video background. However, the previously mentioned works implement this idea in two distinct ways. Nguyen et al. (2021) and Deng et al. (2022) use color images as background and pick a random sub-sequence of a random video for each environment rollout. They adhere to the train-validation split of the Kinetcs400 dataset, using training videos for representation and policy learning and validation videos during evaluation. Zhang et al. (2020); Fu et al. (2021); Wang et al. (2022), according to the official implementations, instead work with gray-scale images and sample a single background video for the train set once during initialization of the environment. They do not sample a new video during the environment reset, thus all training sequences have the same background video. We follow the first approach, as we believe it mimics a more realistic scenario of always changing and colored natural background.

**Occlusions.**  Following (Becker & Neumann, 2022), we render slow-moving disks over the original observations to occlude parts of the observation. The speed of the disks makes memory necessary, as they can occlude relevant aspects for multiple consecutive timesteps.

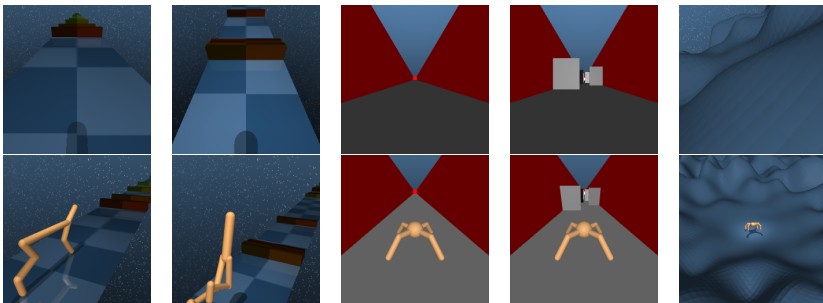

Figure 7: The environments in the Locomotion Suite are (from left to right) Hurdle Cheetah Run, Hurdle Walker Walk / Run, Ant Empty, Ant Walls, and Quadruped Escape. **Upper Row:** Egocentric vision provided to the agent. **Lower Row:** External image for visualization.

## A.2 Locomotion Suite

The 6 tasks in the locomotion suite are `Ant Empty`, `Ant Walls`, `Hurdle Cheetah Run`, `Hurdle Walker Walk`, `Hurdle Walker Run`, and `Quadruped Escape`. Table 2 shows the splits into proprioceptive and non-proprioceptive parts. Fig. 7 displays all environments in the suite.

**Ant.** The Ant tasks build on the locomotion functionality introduced into the DeepMind Control suite by (Tassa et al., 2020). For Ant Empty, we only use an empty corridor, which makes this the easiest task in our locomotion suite. For Ant Walls, we randomly generate walls inside the corridor, and the agent has to avoid those in order to achieve its goal, i.e., running through the corridor as fast as possible.

**Hurdle Cheetah & Walker.** We modified the standard `Cheetah Run`, `Walker Walk`, and `Walker Run` tasks by introducing "hurdles" over which the agent has to step in order to move forward. The hurdles' positions, heights, and colors are reset randomly for each episode, and the agent has to perceive them using egocentric vision. For this vision, we added a camera in the head of the Cheetah and Walker. Note that the hurdle color is not relevant to avoid them and thus introduces irrelevant information that needs to be captured by reconstruction-based approaches.

**Quadruped Escape.** The Quadruped Escape task is readily available in the DeepMind Control Suite. For the egocentric vision, we removed the range-finding sensors from the original observation and added an egocentric camera.

## A.3 *OpenCarbinetDrawer* Environment based on Maniskill2

Both the static and changing background versions of the *OpenCarbinetDrawer* task are based on the mobile manipulation `OpenCabinetDrawer`-task from Maniskill2 (Gu et al., 2023). Both versions use the normalized dense reward provided by the original environment. We disable the rotation of the robot base, as we found this significantly speeds up learning for all considered approaches. This results in a 10 dimensional action space, consisting of the $x$ and $y$ velocities of the base, desired changes for the 7 robot joints, and the gripper. Images are egocentric from the top of the robot base and the proprioception includes the entries from the ManiSkill2 "state dict".

For the constant background variant, we use the "minimal bedroom" scene provided by ManiSkill2. For the changing background variant, we work with the scenes from the Replica Dataset Straub et al. (2019) (specifically: ReplicaCAD_baked_lighting[2]). We pick 80 scenes and create hand-picked offsets to ensure realistic drawer placement. Additionally, we randomly sample RGB values for the ambient lighting, changing the overall appearance of the scenes.

For the depth images we use the depth camera functionality provided by ManiSkill2 and clip to values between 0 and 4 meters.

---

[2]https://huggingface.co/datasets/ai-habitat/ReplicaCAD_baked_lighting/

# B ARCHITECTURE DETAILS, TRAINING, AND BASELINES

We used the same hyperparameters for all experiments based on the DeepMind Control Suite (DMC), i.e., the standard tasks with the different observation types, as well as, the locomotion tasks. For the Maniskill2 (Gu et al., 2023) based *OpenCarbinetDrawer* tasks, we used a slightly larger model and a more conservative update scheme for actor and critic.

## B.1 *Recurrent State Space Model*

We denote the deterministic part of the *RSSM*'s state by $\mathbf{h}_t$ and the stochastic part by $\mathbf{s}_t$. The base-*RSSM* model without parts specific to one of the objectives consists of:

- **Encoders:** $\psi_{\text{obs}}^{(k)}(\mathbf{o}_t)$, where $\psi_{\text{obs}}$ is the convolutional architecture proposed by (Ha & Schmidhuber, 2018) and used by (Hafner et al., 2019; 2020) for image observations. For the low-dimensional proprioception, we used $3 \times 400$ Units fully connected NN with ELU activation for the DMC tasks and a $4 \times 512$ Units fully connected NN with ELU activation for *OpenCarbinetDrawer*.

- **Deterministic Path**: $\mathbf{h}_t = g(\mathbf{z}_{t-1}, \mathbf{a}_{t-1}, \mathbf{h}_{t-1}) = \text{GRU}(\psi_{\text{det}}(\mathbf{z}_{t-1}, \mathbf{a}_{t-1}), \mathbf{h}_{t-1})$ (Cho et al., 2014), where $\psi_{\text{det}}$ is $2 \times 400$ units fully connected NN with ELU activation and the GRU has a memory size of 200 for the DMC tasks. For *OpenCarbinetDrawer* the has $2 \times 512$ units and the GRU a memory size of 400

- **Dynamics Model**: $p(\mathbf{z}_{t+1}|\mathbf{z}_t, \mathbf{a}_t) = \psi_{\text{dyn}}(\mathbf{h}_t)$, where $\psi_{\text{dyn}}$ is a $2 \times 400$ units fully connected NN with ELU activation for the DMC tasks and a $2 \times 512$ units fully connected NN with ELU activation for *OpenCarbinetDrawer*. The network learns the mean and standard deviation of the distribution.

- **Variational Distribution** $q(\mathbf{z}_t|\mathbf{z}_{t-1}, \mathbf{a}_{t-1}, \mathbf{o}_t) = \psi_{\text{var}}\left(\mathbf{h}_t, \text{Concat}\left(\{\psi_{\text{obs}}^{(k)}(\mathbf{o}_t^{(k)})\}_{k=1:K}\right)\right)$, where $\psi_{\text{var}}$ is a $2 \times 400$ units fully connected NN with ELU activation for the DMC tasks and a $2 \times 512$ units fully connected NN with ELU activation for *OpenCarbinetDrawer*. The network learns the mean and standard deviation of the distribution.

- **Reward Predictor** $p(r_t|\mathbf{z}_t)$: $2 \times 128$ units fully connected NN with ELU activation for model-free agents. $3 \times 300$ units fully connected NN with ELU activation for model-based agents. The network only learns the mean of the distribution. The standard deviation is fixed at 1. The model-based agents use a larger reward predictor as they rely on it for learning the policy and the value function. Model-free agents use the reward predictor only for representation learning and work with the ground truth rewards from the replay buffer to learn the critic.

## B.2 OBJECTIVES

**Image Inputs and Augmentation.** For the reconstruction objective, we used images of size $64 \times 64$ pixels as input to the model. For the contrastive objectives, the images are of size $76 \times 76$ pixel image and we used $64 \times 64$ pixel random crops. Cropping is temporally consistent, i.e., we used the same crop for all time steps in a sub-sequence. For evaluation, we took the crop from the center.

**Reconstruction Objectives.** Whenever we reconstructed images we used the up-convolutional architecture proposed by (Ha & Schmidhuber, 2018) and used by (Hafner et al., 2019; 2020). For low-dimensional observations, we used $3 \times 400$ units fully connected NN with ELU activation for the DMC tasks and a $4 \times 512$ Units fully connected NN with ELU activation for *OpenCarbinetDrawer*. In all cases, only the mean is learned. We use a fixed variance of 1 for all image losses and the proprioception for the DMC tasks. For *OpenCarbinetDrawer* we set the variance for the proprioception to $0.04$.

**KL.** For the KL terms in Equation 1 and Equation 3 we follow Hafner et al. (2023) and combine the KL-Balancing technique introduced in Hafner et al. (2021) with the *free-nats regularization* used in Hafner et al. (2019; 2020). Following Hafner et al. (2021) we use a balancing factor of $0.8$. We give the algorithm 1 free nat for the DeepMind Control Suite and the Locomotion Suite tasks and 3 for *OpenCarbinetDrawer*.

Table 3: Hyperparameters used for policy learning with the Soft Actor-Critic.

| Hyperparameter | DMC and Locomotion | *OpenCarbinetDrawer* |
|---|---|---|
| Actor Hidden Layers | $3 \times 1,024$ Units | $3 \times 1,024$ Units |
| Actor Activation | ELU | ELU |
| Critic Hidden Layers | $3 \times 1,024$ Units | $3 \times 1,024$ Units |
| Critic Activation | ELU | ELU |
| Discount | 0.99 | 0.85 |
| Actor Learning Rate | 0.001 | 0.0003 |
| Actor Gradient Clip Norm | 10 | 10 |
| Critic Learning Rate | 0.001 | 0.0003 |
| Critic Gradient Clip Norm | 100 | 100 |
| Target Critic Decay | 0.995 | 0.995 |
| Target Critic Update Interval | 1 | 1 |
| $\alpha$ learning rate | 0.001 | 0.0003 |
| initial $\alpha$ | 0.1 | 1.0 |
| target entropy | - action dim | - action dim |

**Contrastive Variational Objective.** The score function for the contrastive variational objective is given as

$$f_v^{(k)}(\mathbf{o}_t^{(k)}, \mathbf{z}_t) = \exp\left(\frac{1}{\lambda}\rho_o\left(\psi_{\mathrm{obs}}^{(k)}(\mathbf{o}_t)\right)^T \rho_z(\mathbf{z}_t)\right),$$

where $\psi_{\mathrm{obs}}^{(k)}$ is the *RSSM*'s encoder and $\lambda$ is a learnable inverse temperature parameter. $\rho_o$ and $\rho_z$ are projections that project the embedded observation and latent state to the same dimension, i.e., $50$. $\rho_o$ is only a single linear layer while $\rho_z$ is a $2 \times 256$ fully connected NN with ELU activation. Both use LayerNorm (Ba et al., 2016) at the output.

**Contrastive Predictive Objective.** The score function of the contrastive predictive objective looks similar to the one of the contrastive variational objective. The only difference is that the latent state is forwarded in time using the *RSSMs* transition model to account for the predictive nature of the objective,

$$f_p^{(k)}(\mathbf{o}_{t+1}^{(k)}, \mathbf{z}_t) = \exp\left(\frac{1}{\lambda}\rho_o\left(\psi_{\mathrm{obs}}^{(k)}(\mathbf{o}_{t+1})\right)^T \rho_z(\phi_{\mathrm{dyn}}(g(\mathbf{z}_t, \cdot)))\right).$$

We use the same projections as in the contrastive variational case.

Following Srivastava et al. (2021) we scale the KL term using a factor of $\beta = 0.001$ and parameterize the inverse dynamics predictor as a $2 \times 128$ unit fully connected NN with ELU activations.

**Optimizer.** We used Adam Kingma & Ba (2015) with $\alpha = 3 \times 10^{-4}$, $\beta_1 = 0.99, \beta_2 = 0.9$ and $\varepsilon = 10^{-8}$ for all losses. We clip gradients if the norm exceeds 10.

## B.3 SOFT ACTOR CRITIC

Table 3 lists the hyperparameters used for model-free RL with SAC Haarnoja et al. (2018).

We collected 5 initial sequences at random. During training, we update the *RSSM*, critic, and actor in an alternating fashion for $d$ steps before collecting a new sequence by directly sampling from the maximum entropy policy. Here, $d$ is set to be half of the environment steps collected per sequence (after accounting for potential action repeats) for DMC tasks and 50 for *OpenCarbinetDrawer*. Each step uses 32 subsequences of length 32, uniformly sampled from all prior experience.

## B.4 LATENT IMAGINATION

Table 4 lists the hyperparameters used for model-based RL with latent imagination. They follow to a large extent those used in Hafner et al. (2020; 2021).

Table 4: Hyperparameters used for policy learning with *Latent Imagination.*

| Hyperparameter | Value |
|---|---|
| Actor Hidden Layers | $3 \times 300$ Units |
| Actor Activation | ELU |
| Critic Hidden Layers | $3 \times 300$ Units |
| Critic Activation | ELU |
| Discount | 0.99 |
| Actor Learning Rate | $8 \times 10^{-5}$ |
| Actor Gradient Clip Norm | 100 |
| Value Function Learning Rate | $8 \times 10^{-5}$ |
| Value Gradient Clip Norm | 100 |
| Slow Value Decay | 0.98 |
| Slow Value Update Interval | 1 |
| Slow Value Regularizer | 1 |
| Imagination Horizon | 15 |
| Return lambda | 0.95 |

We collected 5 initial sequences at random. During training, we update the *RSSM*, value function, and actor in an alternating fashion for 100 steps before collecting a new sequence. Each step uses 50 subsequences of length 50, uniformly sampled from all prior experience. For collecting new data, we use constant Gaussian exploration noise with $\sigma = 0.3$.

### B.5 BASELINES.

For *Dreamer-v3* (Hafner et al., 2023) we use the raw reward curve data provided with the official implementation[3]. For *DreamerPro* (Deng et al., 2022)[4], *Task Informed Abstractions* (Fu et al., 2021)[5], *Deep Bisumlation for Control* Zhang et al. (2020)[6], *DenoisedMDP* (Wang et al., 2022)[7] and *DrQ-v2* (Yarats et al., 2022)[8] we use the official implementations provided by the respective authors.

**DrQ-(I+P)** builds on the official implementation and uses a separate encoder for the proprioception whose output is concatenated to the image encoders' output and trained using the critics' gradients.

**Differences between Model-Based *Img-Only(R)* and Dreamer-v1(Hafner et al., 2020).** *Img-Only(R)* differs from the original Dreamer (Dreamer-v1) (Hafner et al., 2020) in using the KL-balancing introduced in (Hafner et al., 2021) and in regularizing the value function towards its own exponential moving average, as introduced in (Hafner et al., 2023). See Appendix B for all our training details and hyperparameters.

There are considerable differences between the contrastive version of Dreamer-v1(Hafner et al., 2020) and *Img-Only(CV)*, in particular regarding the exact form of the mutual information estimation and the use of image augmentations.

**Differences between Model-Free *Img-Only(CPC)*, *Joint(CPC+CPC)* and the approach of Srivastava et al. (2021).** The main difference is that (Srivastava et al., 2021) includes the critic's gradients when updating the representation while in our setting no gradients flow from the actor or the critic to the representation. Furthermore, we did not include the inverse dynamics objective used by Srivastava et al. (2021) as we did not find it to be helpful. Additionally, we adapted some hyper-

---

[3]https://github.com/danijar/dreamerv3/blob/main/scores/data/dmcvision_dreamerv3.json.gz
[4]https://github.com/fdeng18/dreamer-pro
[5]https://github.com/kyonofx/tia/
[6]https://github.com/facebookresearch/deep_bisim4control/
[7]https://github.com/facebookresearch/denoised_mdp
[8]https://github.com/facebookresearch/drqv2

parameters to match those of our other approaches. The results are based on our implementation, not the official implementation of (Srivastava et al., 2021).

**Why DBC, TIA, and DenoisedMDP Fail in Our Setting.** *Deep Bisimulation for Control (DBC)* (Zhang et al., 2020), *Task Informed Abstractions (TIA)* (Fu et al., 2021), and *DenoisedMDP* (Wang et al., 2022) fail to perform well in the more challenging natural video background setting introduced by Nguyen et al. (2021) and Deng et al. (2022). As described in Section A.1, there are differences in video selection and sampling.

The failure of *DBC* in this setting is in line with the findings of Nguyen et al. (2021).

*TIA* and *DenoisedMDP* factorize the latent variable into 2 distinct parts and formulate loss functions that force one part to focus on task-relevant aspects and the other on task-irrelevant aspects. However, the part responsible for the task-irrelevant aspects still has to model those explicitly. In the more complicated setting with randomly sampled, colored background videos, the *TIA* and *DenoisedMDP* world models underfit and thus fail to learn a good representation or policy. Contrastive approaches, such as our approach and DreamerPro (Deng et al., 2022), do not struggle with this issue, as they do not have to model task-irrelevant aspects but can learn to ignore them.

### B.6 COMPUTE RESOURCES

Training a single agent for any of the DMC tasks takes between 8 and 12 hours on a single GPU (Nvidia V100 or A100), depending on which representation learning approach and RL paradigm we use. For *OpenCabinetDrawer* one run takes about 30 hours due to the more complex simulation. Approaches using a contrastive loss for the image are slightly faster than those that reconstruct the image as they do not have to run the relatively large up-convolutional image decoder. The model-free agents train slightly faster than the model-based ones, as the model-based ones have to predict several steps into the future for latent imagination. Especially propagating gradient back through this unrolling is relatively costly compared to a SAC update. Including all baselines, we trained about $4,000$ agents for the final evaluation. Also including preliminary experiments, we estimate the total compute resources invested in this work to be about $70,000$ GPU hours.

## C  COMPLETE RESULTS

The following pages list the aggregated results and performance profiles for all tasks, representation-learning approaches, and both model-free and model-based RL. We compute inter-quartile means and stratified bootstrapped confidence intervals, as well as the performance profiles according to the recommendations of Agarwal et al. (2021) using the provided library[9]. For each task in the suites we ran 5 seeds per method, i.e., the results for *Standard Images*, *Video Backgrounds*, and *Occlusions* are aggregated over 35 runs and those for *Locomotion* over 30 runs. For *OpenCabinetDrawer* we run 20 seeds per method. Fig. 8 lists the aggregated results for all model-free agents on the DeepMind Control (DMC) Suite tasks and Fig. 10 lists the corresponding performance profiles. Fig. 9 lists the aggregated results for all model-based agents on the DeepMind Control Suite tasks and Fig. 11 lists the corresponding performance profiles. Fig. 12 shows aggregated results and performance profiles for the *Locomotion* suite. Fig. 13 shows the results for the *OpenCarbinetDrawer* with color images and Fig. 14 shows the results for the *OpenCarbinetDrawer* with depth images. We also list the per-environment results for the remaining environments:

- Fig. 15: Model-free agents on DMC tasks with *Standard Images*
- Fig. 16: Model-free agents on DMC tasks with *Video Background*.
- Fig. 17: Model-free agents on DMC tasks with *Occlusions*.
- Fig. 18: Model-based agents on DMC tasks with *Standard Images*.
- Fig. 19: Model-based agents on DMC tasks with *Video Background*.
- Fig. 20: Model-based agents on DMC tasks with *Occlusions*.
- Fig. 21: Model-free agents on *Locomotion* tasks.

---

[9]https://github.com/google-research/rliable

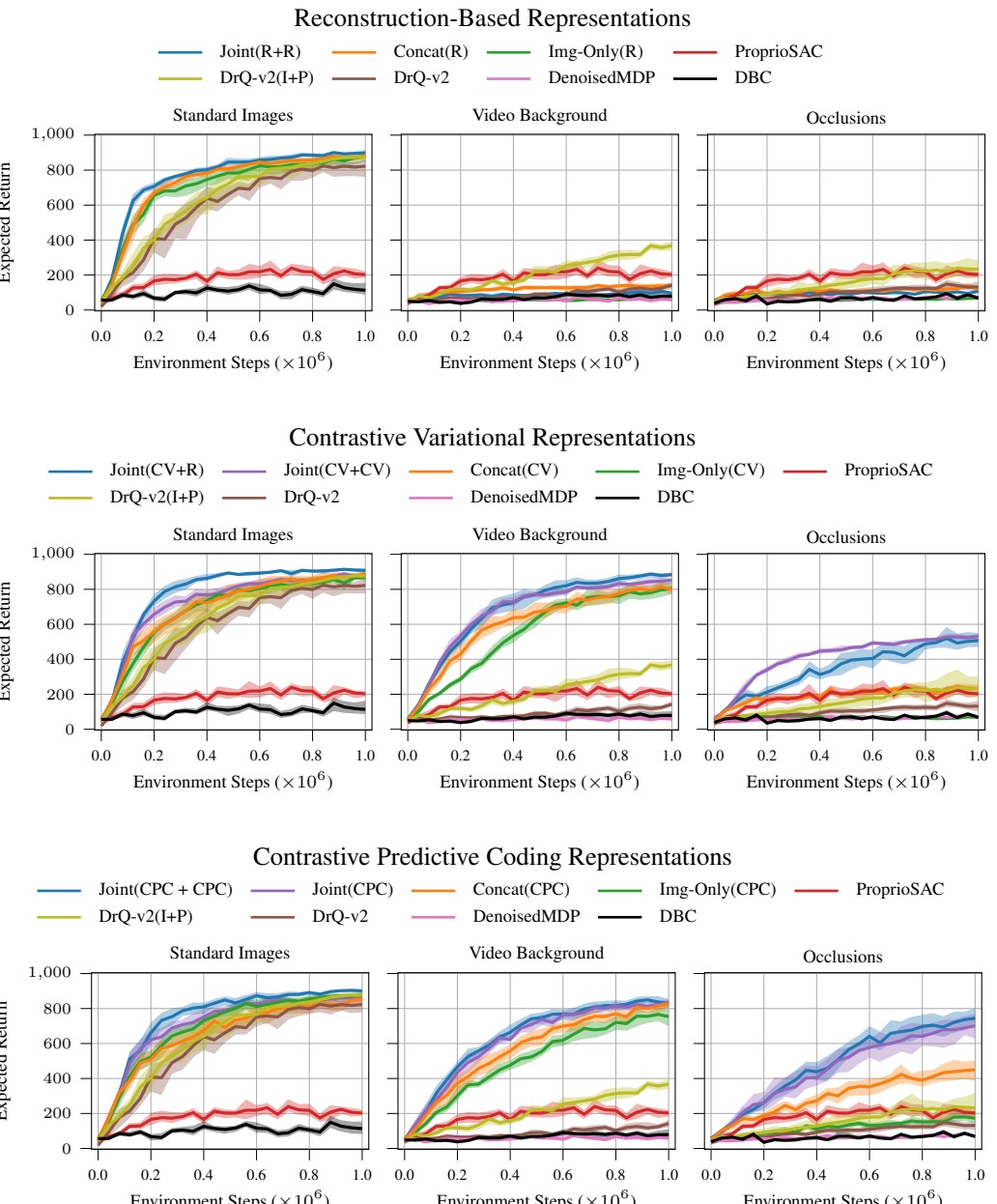

Figure 8: Aggregated results for all **model-free** agents on the DeepMind Control Suite environments with *Standard Images*, *Video Background*, and *Occlusions*. As expected, reconstruction-based approaches do not work on *Video Background* and *Occlusions*. Out of all approaches considered in this work *Joint(CV+R)* achieves the highest performance on *Video Background* and *Joint(CPC+R)* achieves the highest performance on *Occlusions*.

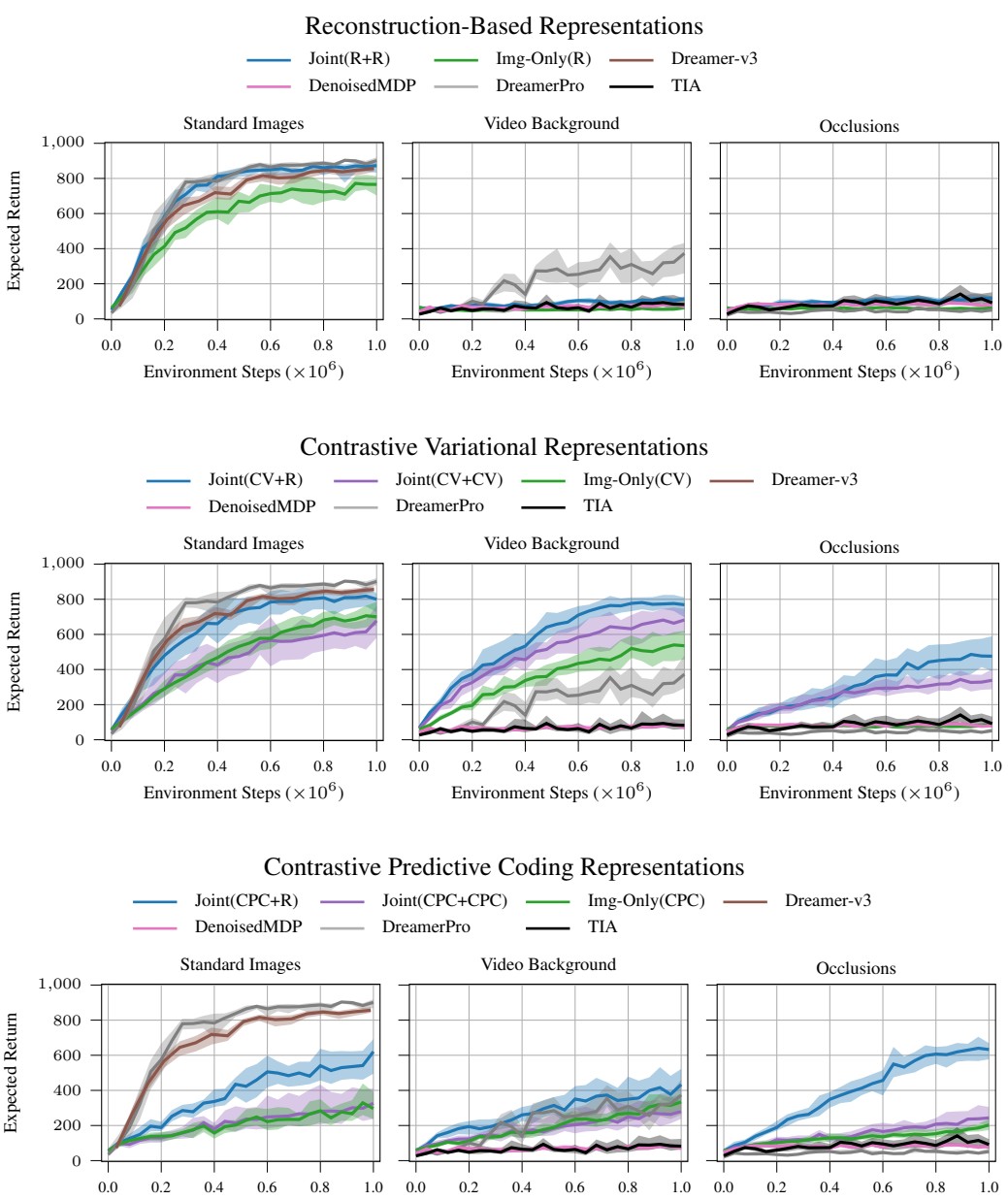

Figure 9: Aggregated results for all **model-based** agents on the DeepMind Control Suite environments with *Standard Images*, *Video Background*, and *Occlusions*. Compared to their model-free counterparts (Fig. 8), model-based agents perform worse, except if a reconstruction-based representation is used. Yet, the performance gap is larger for image-only and fully contrastive approaches. Especially *Joint(CV+R)* still achieves high performance on *Video Background*, almost matching the performance of *Dreamer-v3* on *Standard Images*. This further highlights the benefits of using joint representations with a mixed objective, which can significantly improve over tailored approaches such as *DreamerPro*.

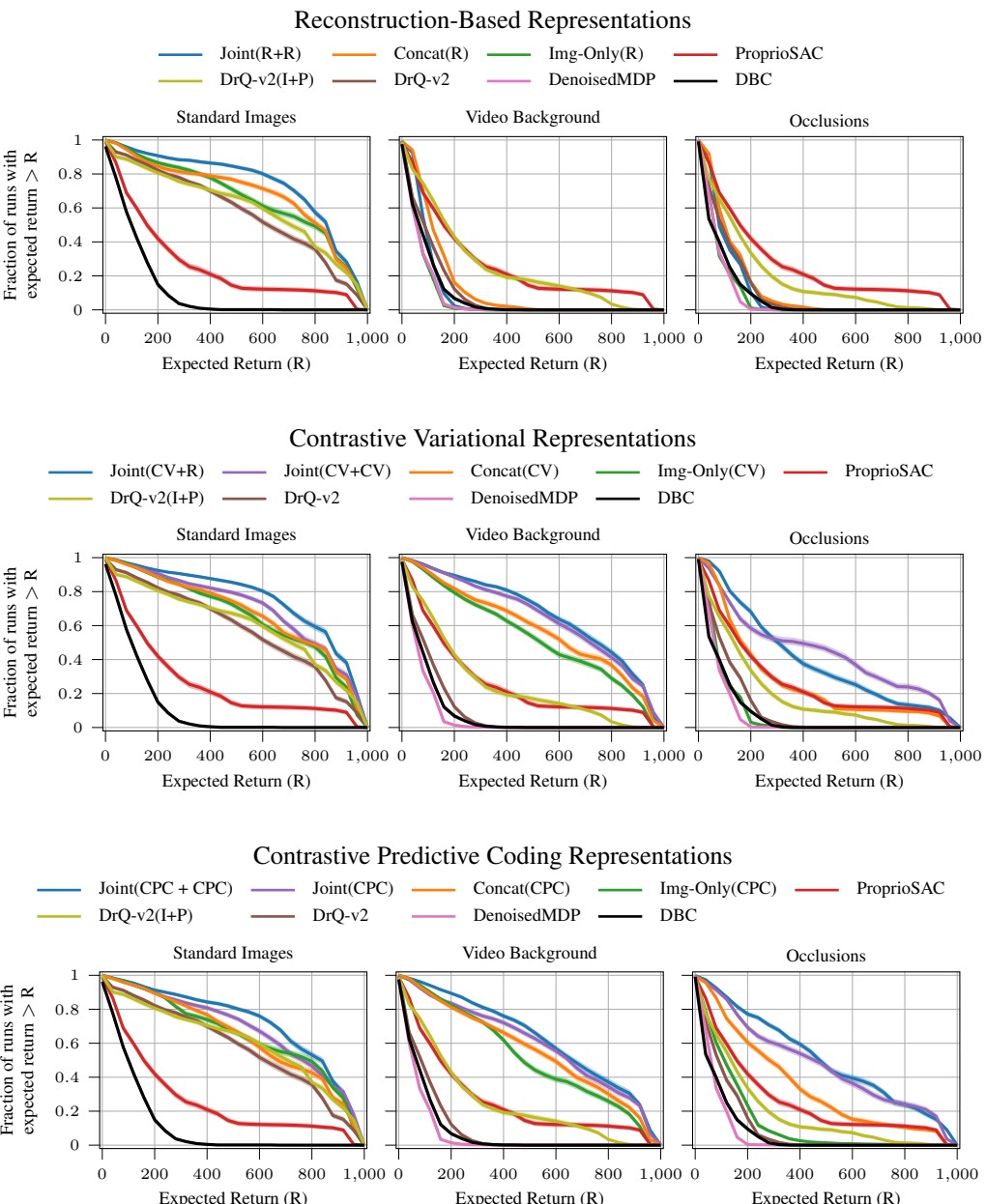

Figure 10: Performance profiles for all **model-free** agents on the DeepMind Control Suite environments with *Standard Images*, *Video Background*, and *Occlusions*. They indicate that performance is largely consistent across the environments. The sole exception is *Joint(CV+R)* and *Joint(CV+CV)* on *Occlusions*. Here, the former fails for `Ball-in-Cup Catch` and *Cartpole Swingup*, while the latter underperforms for *Cheetah Run* (c.f. Fig. 17).

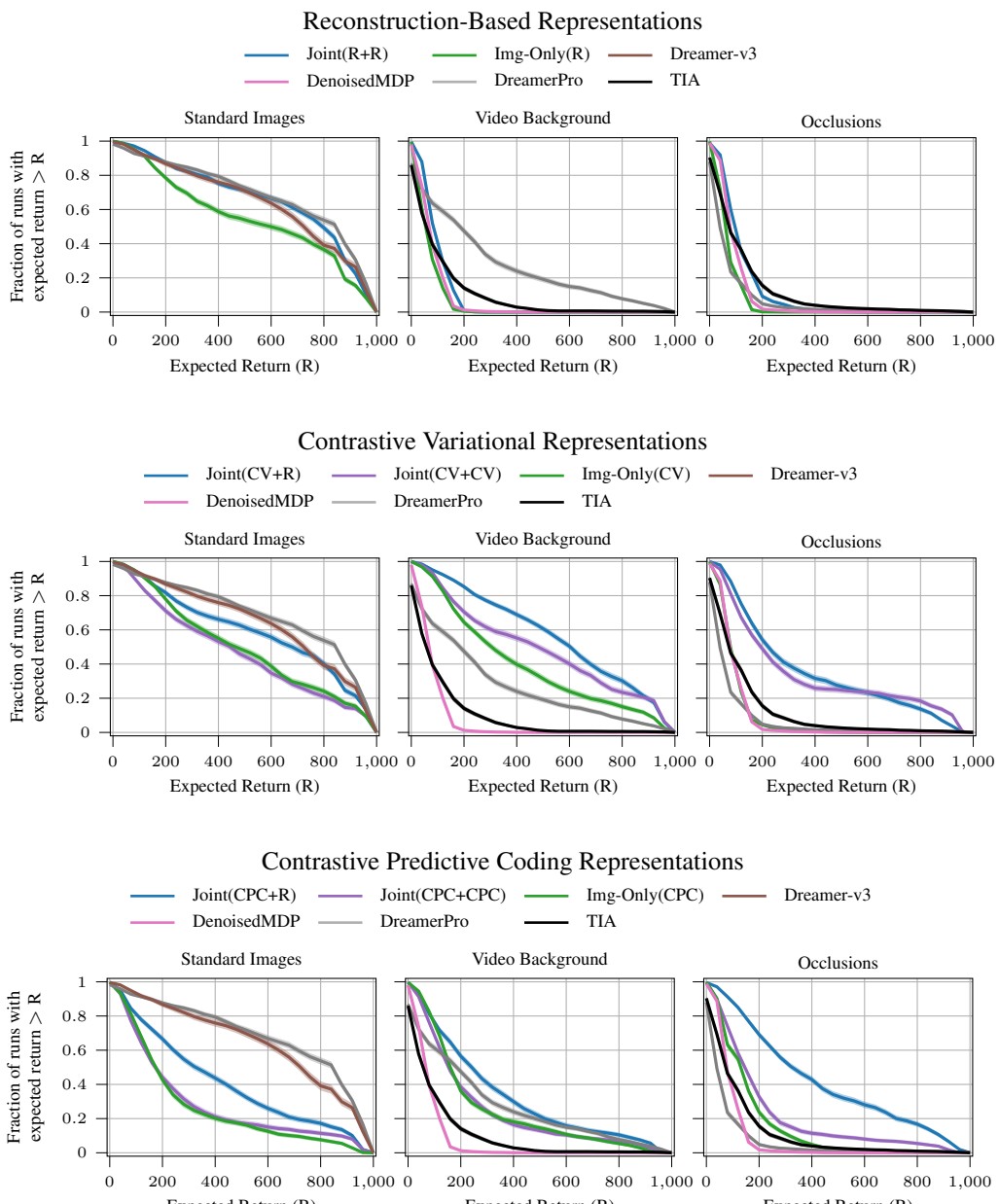

Figure 11: Performance profiles for all **model-based** agents on the DeepMind Control Suite environments with *Standard Images*, *Video Background*, and *Occlusions*. They indicate that performance is largely consistent across the environments.

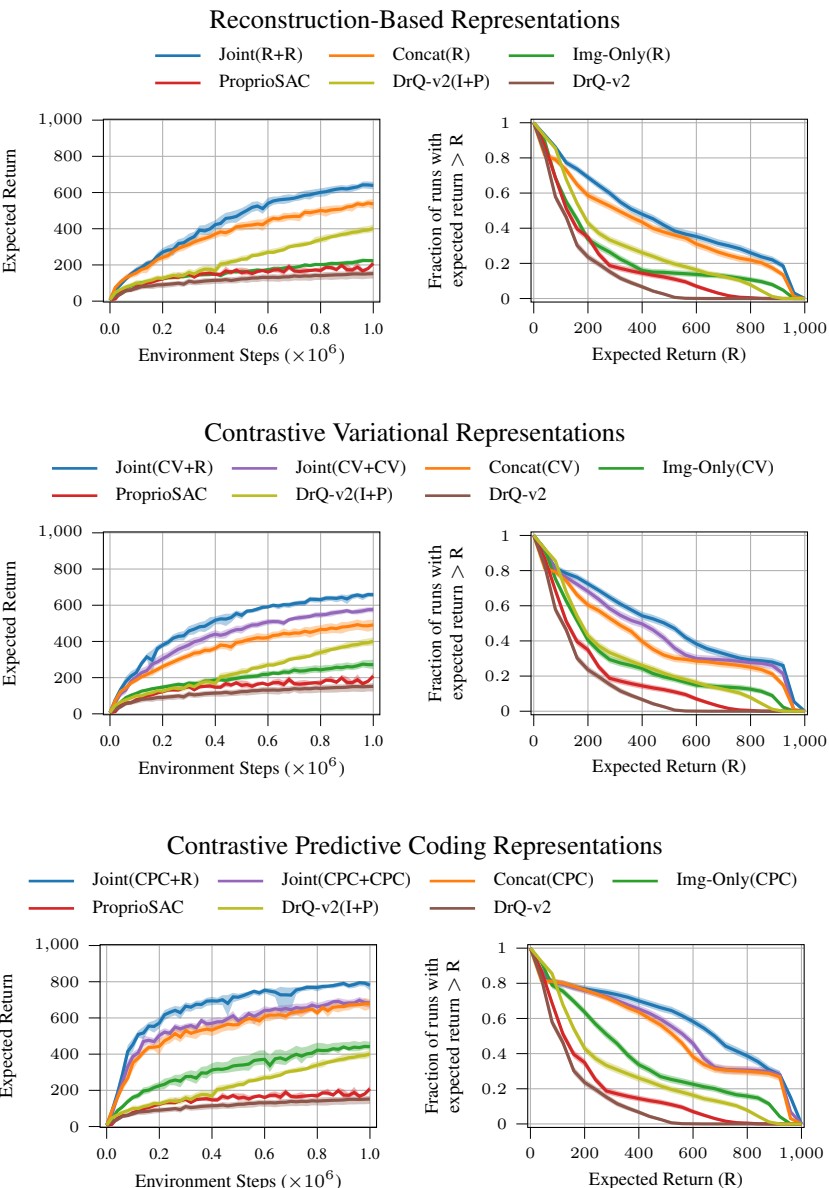

Figure 12: Aggregated results and performance profiles for all **model-free** agents on *Locomotion* environments. Both contrastive approaches outperform reconstruction. Fig. 21 shows that the performance difference is larger in environments with randomly colored obstacles (`Hurdle Cheetah Run`, `Hurdle Walker Walk`, `Hurdle Walker Run`. The color is not relevant to avoid the obstacles but seems to hinder reconstruction.

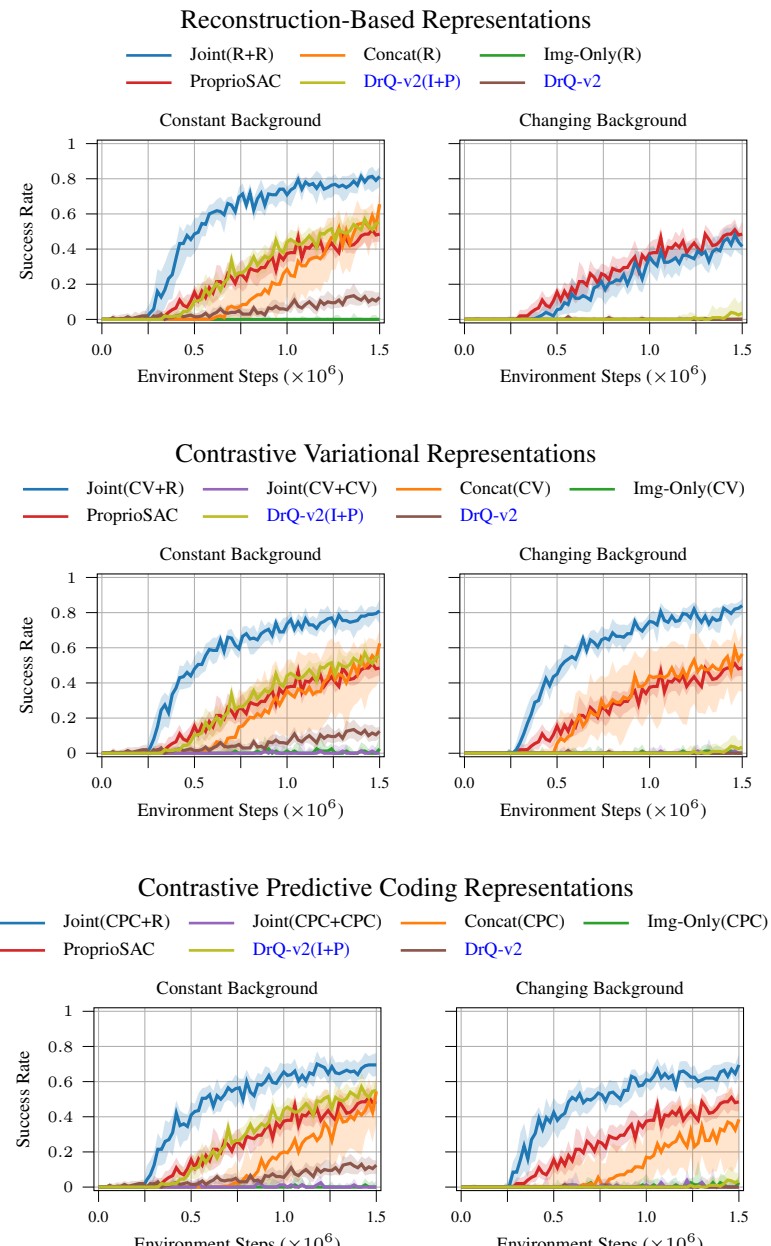

Figure 13: Results for model-free agents on both *OpenCabientDrawer* tasks with color images.. *Joint(CV+R)* achieves the highest performance among all considered methods on both tasks. If the background remains constant, *Joint(R+R)* performs similarly but its performance significantly deteriorates for changing backgrounds. No *Img-Only* approach, *Joint(CV+CV)*, or *Joint(CPC+CPC)* learns a policy that achieves any success and *Concat* performs on par with the *Proprio-Only*. Concatenating a reconstruction-based representation to the proprioception *Concat(R)* even breaks the approach on changing backgrounds. Likely, from the policy's viewpoint, the representation appears to be noise and hinders decision-making.

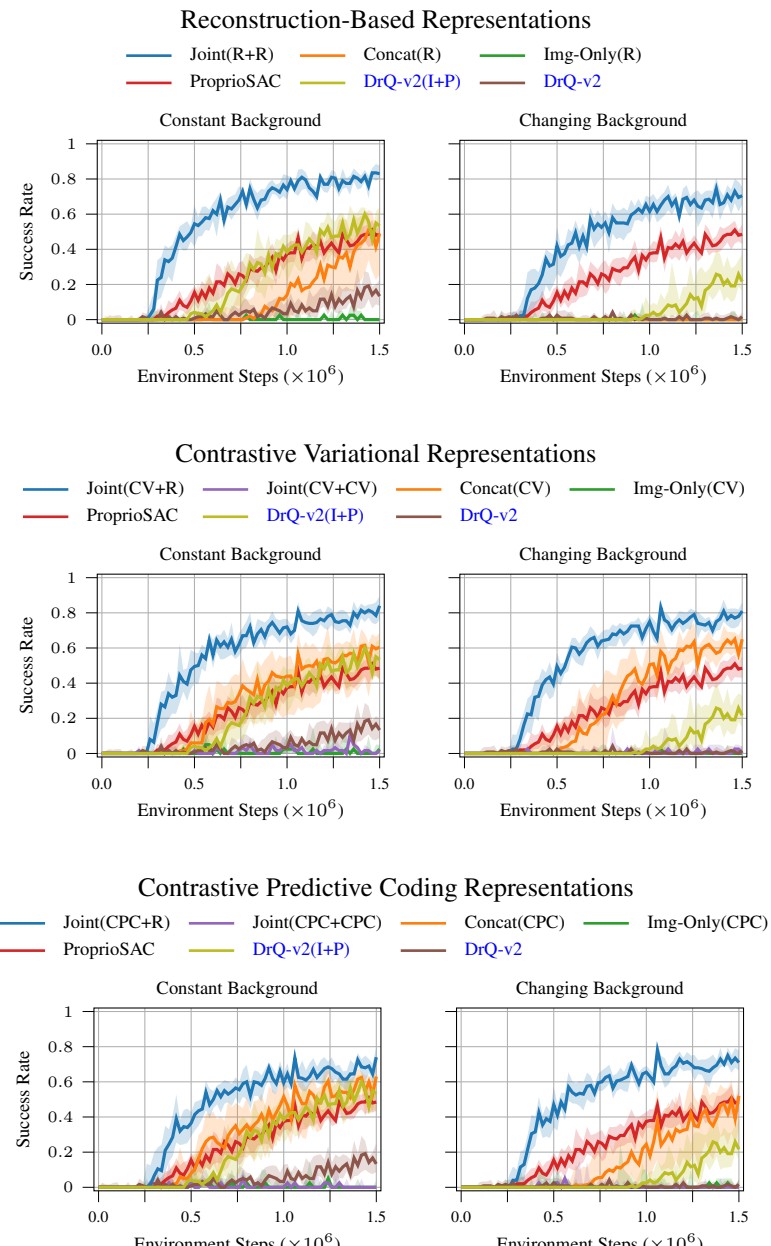

Figure 14: Results for model-free agents on both *OpenCabientDrawer* tasks with depth images. For constant background, the results are similar to the environments with color images (Fig. 13). *Joint(R+R)* and several baselines suffer less from adding the changing background than in the color image setting but are still outperformed by *Joint(CV+R)* and *Joint(CPC+R)*, which retain their performance. These results show how joint representations can help not only with standard color images but also in combination with depth information.

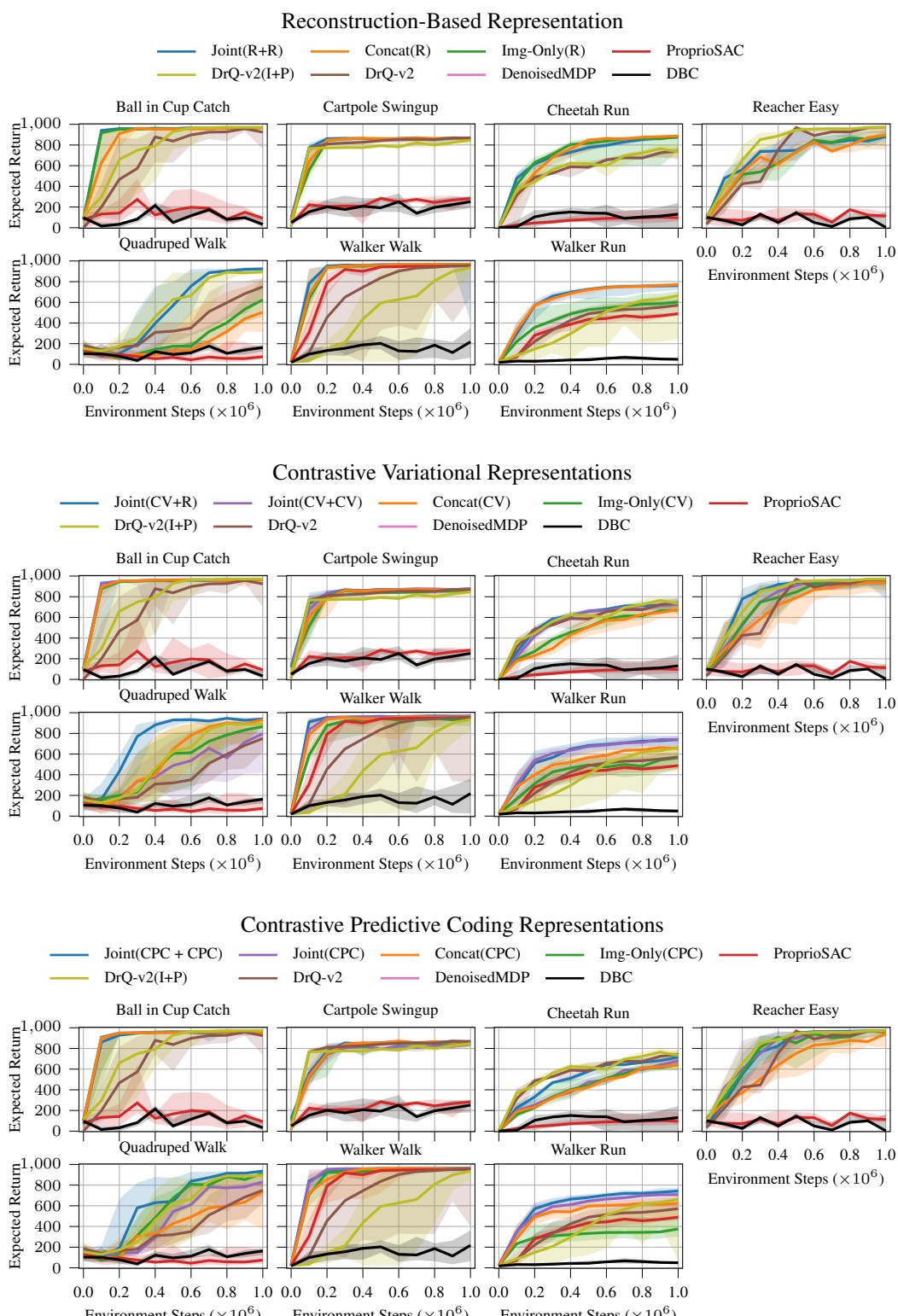

Figure 15: Per environment results for model-free agents on the DeepMind Control Suite with *Standard Images*.

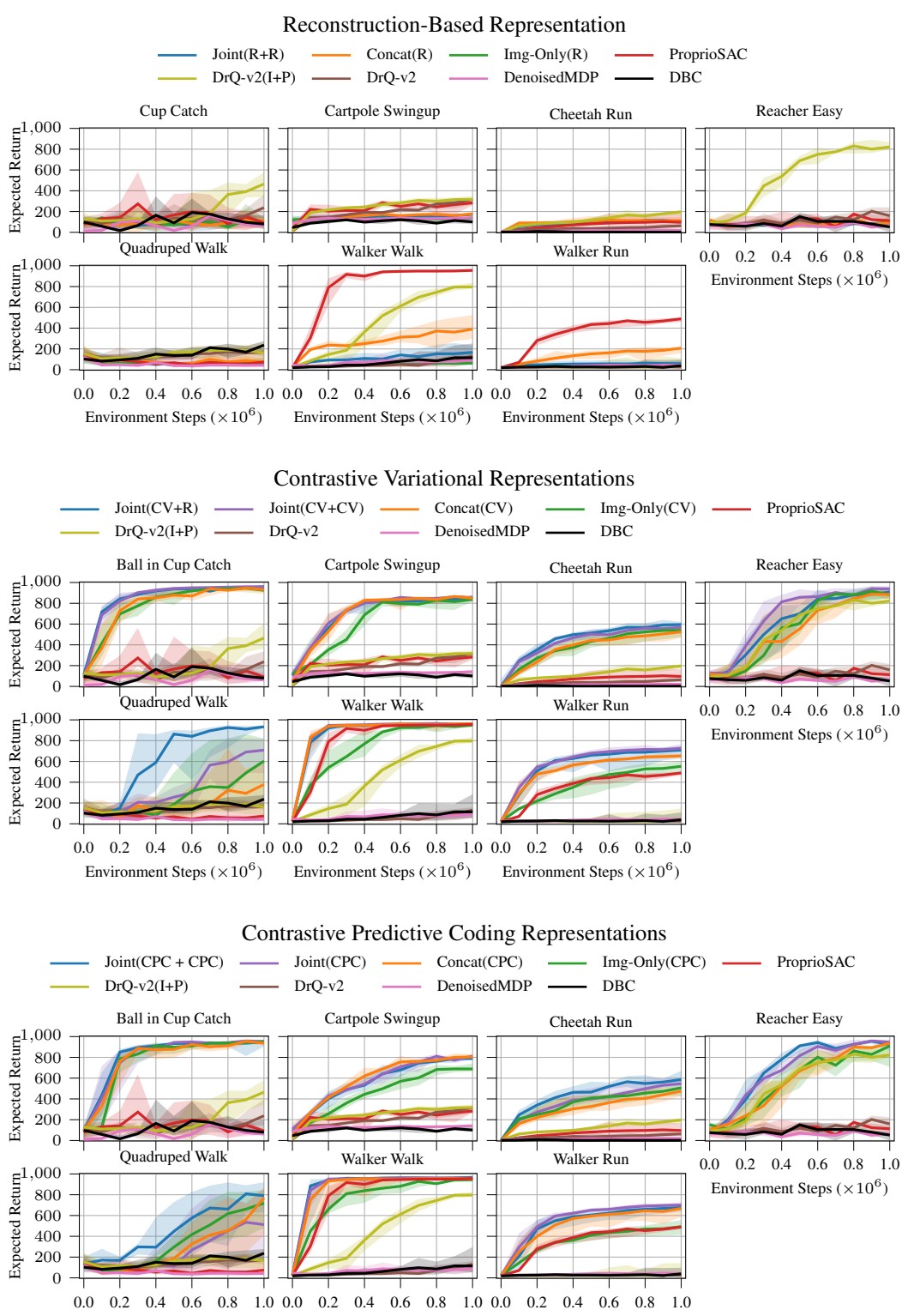

Figure 16: Per environment results for model-free agents on the DeepMind Control Suite with *Video Background*.

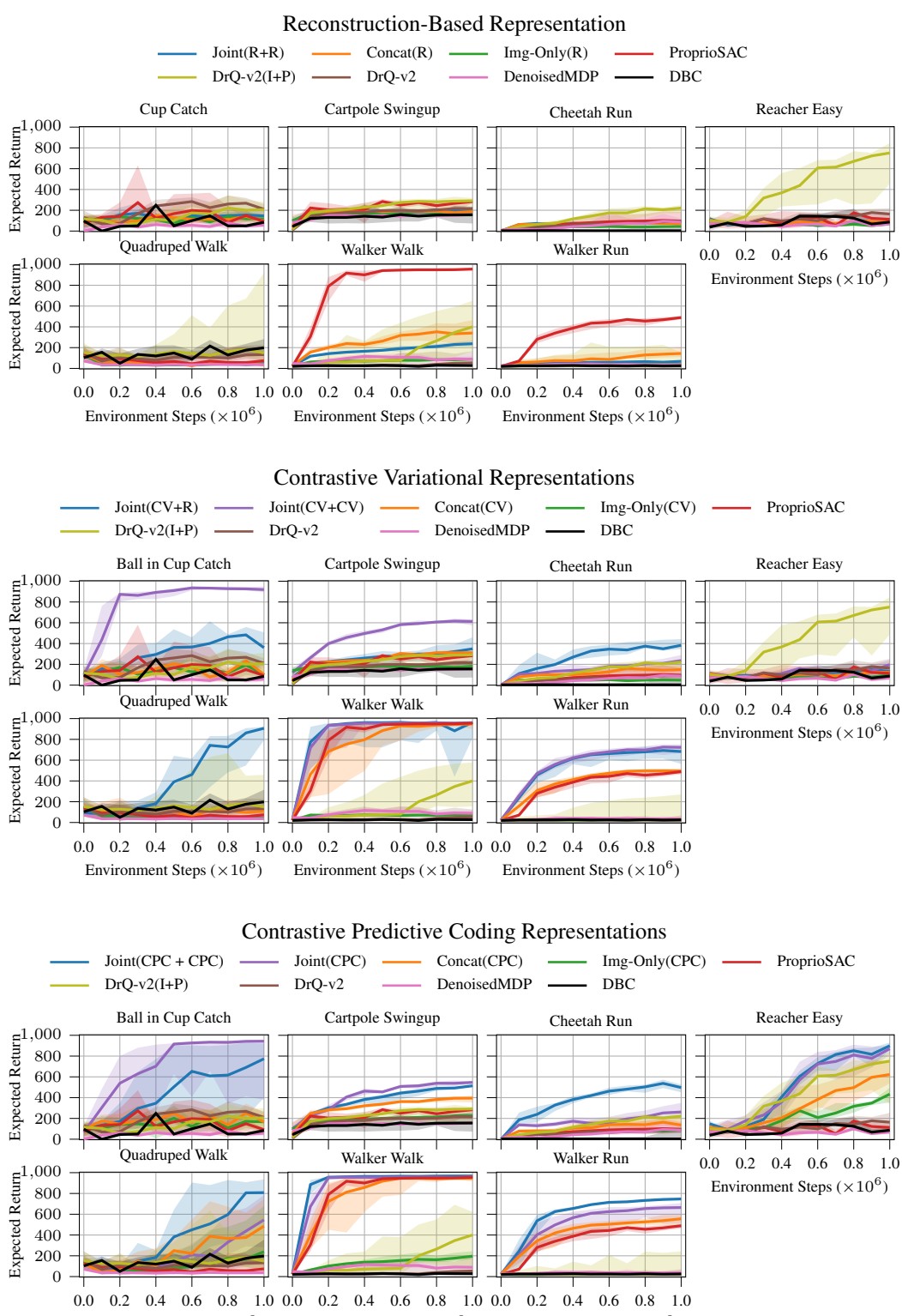

Figure 17: Per environment results for model-free agents on the DeepMind Control Suite with *Occlusions*.

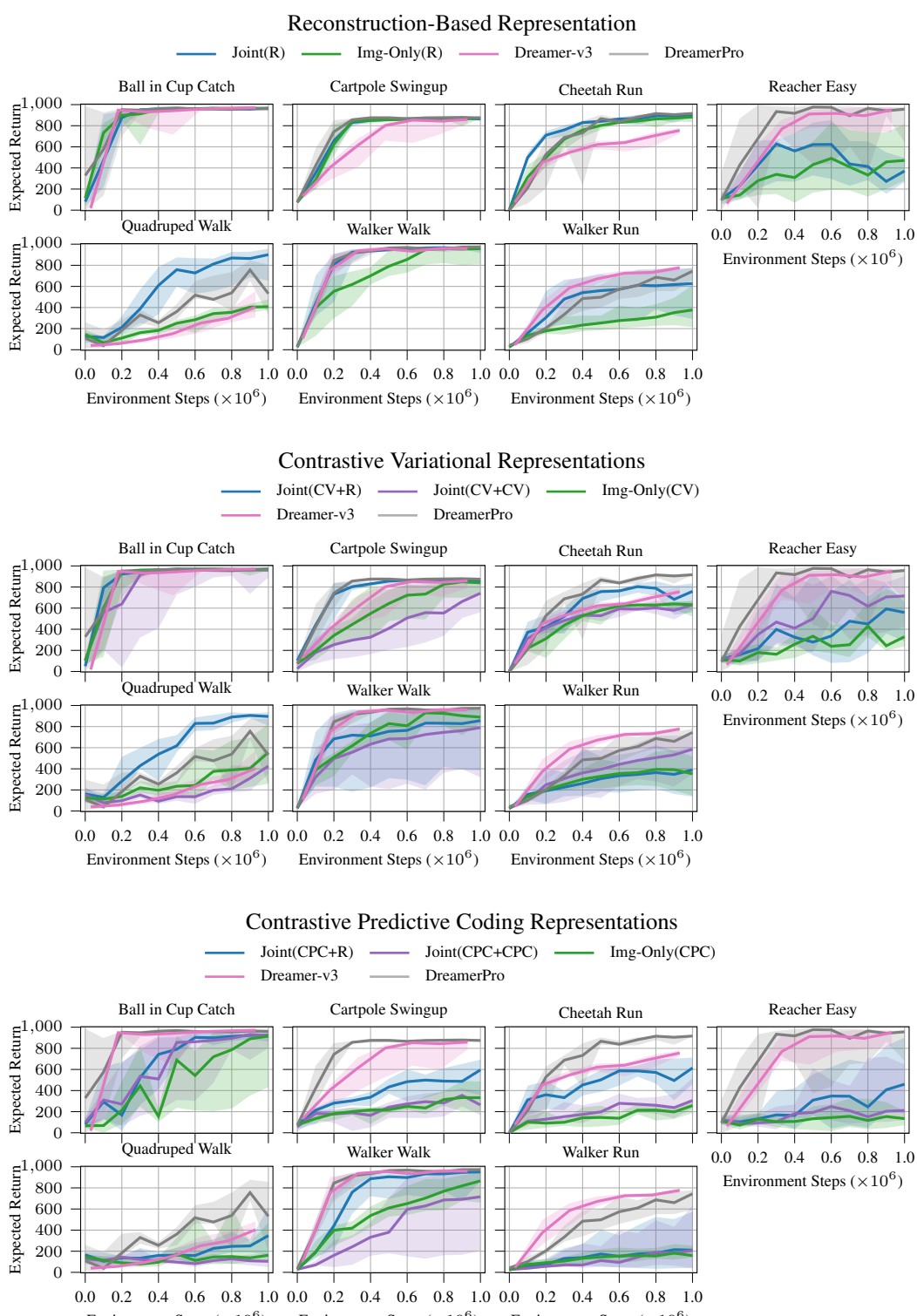

Figure 18: Per environment results for model-based agents on the DeepMind Control Suite with *Standard Images*.

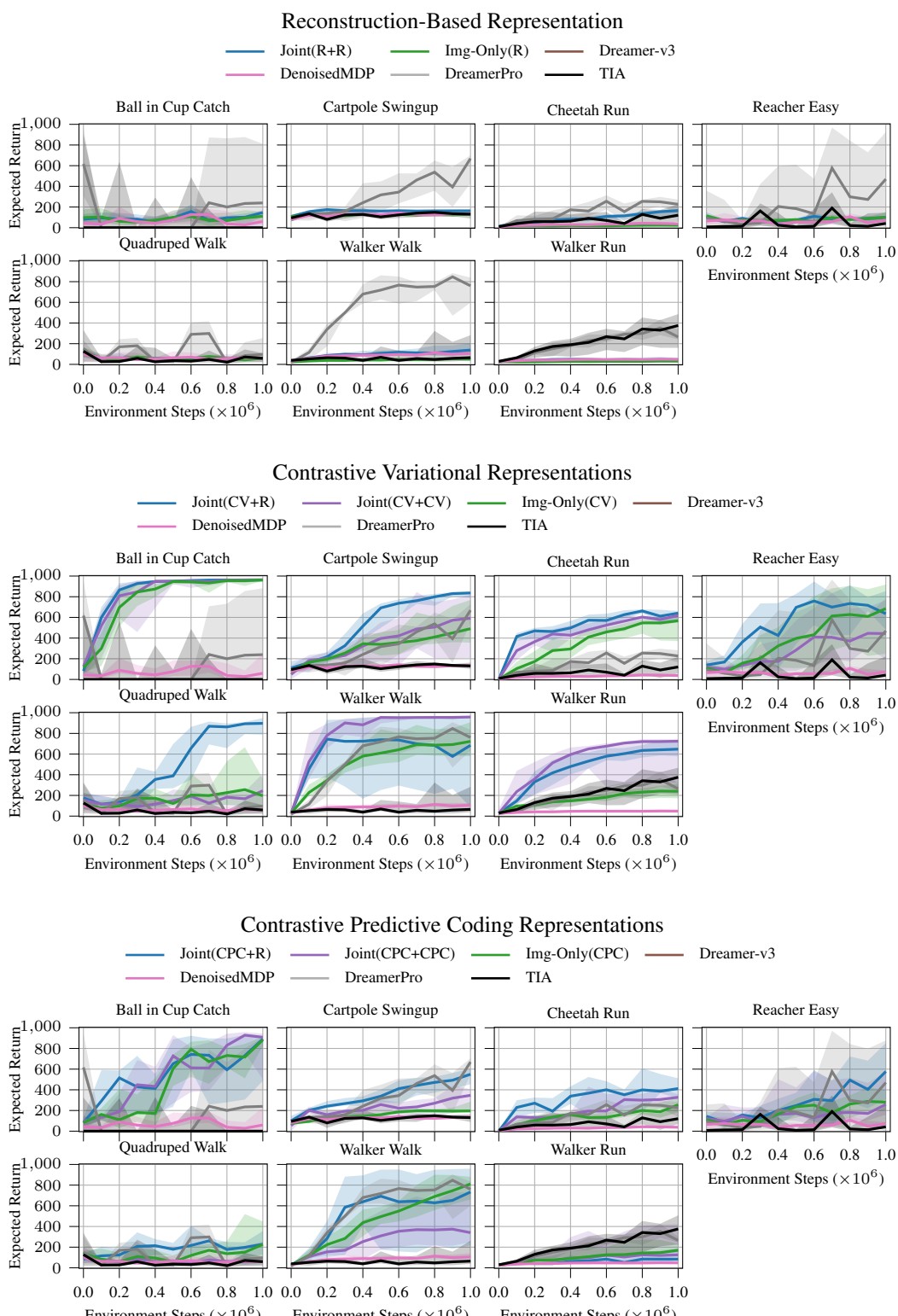

Figure 19: Per environment results for model-based agents on the DeepMind Control Suite with *Video Background*.

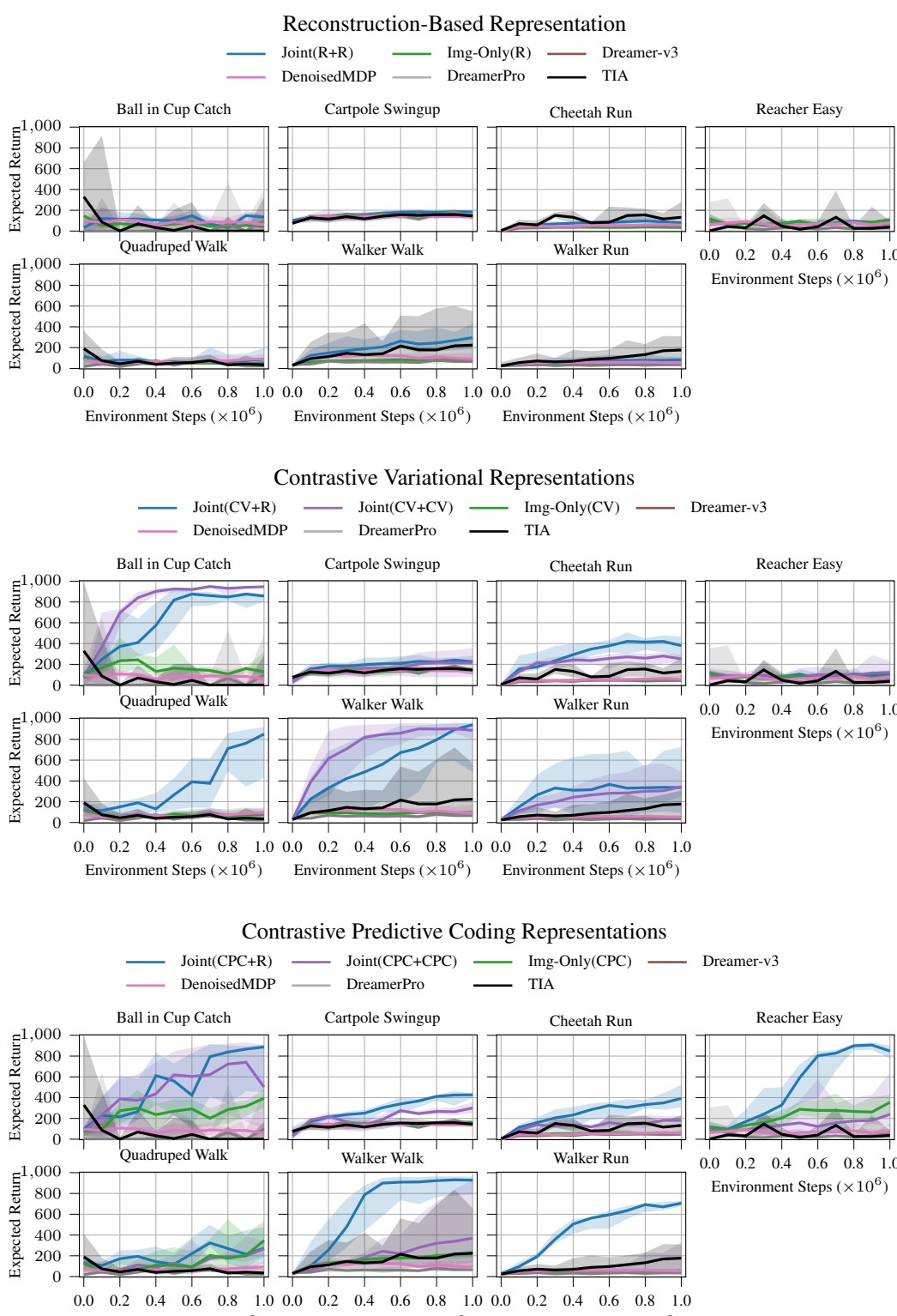

Figure 20: Per environment results for model-based agents on the DeepMind Control Suite with *Occlusions*.

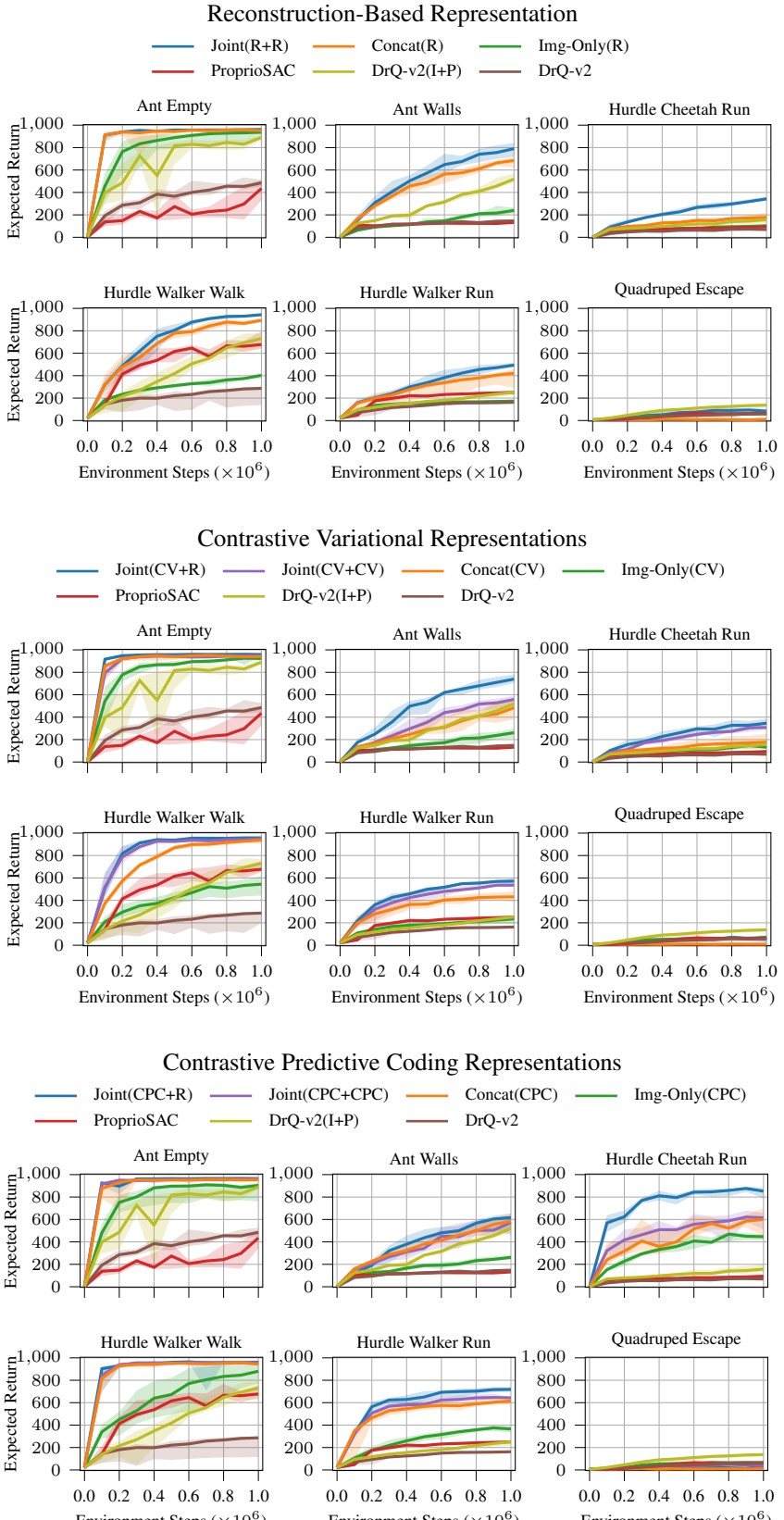

Figure 21: Per environment results for model-free agents on *Locomotion*.

