# OpenReview forum: "Joint Representations for Reinforcement Learning with Multiple Sensors"
_ICLR.cc/2024/Conference — Submitted to ICLR 2024_

### Official Review · Reviewer_VgNz · 2023-10-26

**Soundness:** 3 good
**Presentation:** 3 good
**Contribution:** 2 fair
**Rating:** 5
**Confidence:** 5

**Summary:**

This paper proposes representation learning in reinforcement learning using multi-modal data sources. It aims to enhance representation learning by constructing Recurrent State Space Models tailored with specific objectives for each modality. This work's contribution lies in optimizing the integration of low-dimensional modalities (like proprioception) with high-dimensional, noisy modalities (such as images) to enhance representation learning for RL. The paper suggests employing a reconstruction loss for proprioception data and a contrastive loss for image observations. This work performs experiments on a modified version of the DeepMind Control Suite (DMC) and Mujoco tasks such as Cheetah Run and OpenCabinetDrawer task from ManiSkill2. The results underscore that utilizing a combined representation with appropriate loss functions can improve the performance of RL-based methods.

**Strengths:**

- This framework introduces a clear joint training framework for multi-modal reinforcement learning. It employs reconstruction loss proprioception data and contrastive losses for noisy high-dimensional inputs, such as images. The method is straightforward in both comprehension and implementation.
- Extensive testing on various benchmarks as tasks and baseliens including model-free and model-based RL baselines.

**Weaknesses:**

- Firstly, if the joint representations (adding proprio and images) improves performance over learning an image-only or proprio-only representation, I do not find this surprising. It makes sense that adding more informations improves the performance.
- Secondly, adding different losses for each modality, contrastive for images and reconstruction for proprioception, as the central contribution of this work is weak. What can however make this paper a stronger contribution is pursuing other sensor modalities (depth images, surface normals, segmentations, etc) and then exploring various appropriate losses there. As it stands, the paper is only applying a typical reconstruction-based loss to the proprio and contrastive to the image, which are the current norms in the field - no exciting surprise!
- Thirdly, even though this work provides extensive experiment results, in many figures, the methods showing the effect of each loss, for instance `Joint(R+R)` and `Joint(CV+R)`, the performance of these seem to be on-par with each other e.g. `Figure 8, Figure 2, Figure 3` etc.
- Even though its nice the huge amount of analysis performed in this work, the concluding story is very hard to digest, specially since there are many different acronyms to their proposed method such as `Joint(R+R), Joint(CPC+R), Joint(CV+R)`. In addition, some baselines are missing for some tasks (e.g. `Figure 5`) and it makes drawing a final conclusion hard.
- I also disagree with the following statement made in the paper ` While many self-supervised representation ...  neglect other available information, such as robot proprioception`.  Adding proprioceptive state observations along with images is not novel in the field (especially in robotics).

**Questions:**

- Except for combining multi-modal inputs with appropriate losses per each, are there any other interesting take-away observations from this work?
- Is there a reason many of the model-based baselines are missing for the `OpenCabinetDrawer` task?

---

> ### Author Response · Authors · 2023-11-17
>
> We thank the reviewer for their time and effort and for acknowledging our contributions of a clear joint training framework for multi-modal RL and extensive validation on various benchmarks. The revised version of our manuscript more prominently highlights our key contributions and takeaways. It also includes several clarifications in response to the raised concerns. In the following we address these individual concerns in more detail
>
> > Firstly, if the joint representations (adding proprio and images) improves performance over learning an image-only or proprio-only representation, I do not find this surprising. [...]
>
> > Secondly, adding different losses for each modality, contrastive for images and reconstruction for proprioception, as the central contribution of this work is weak. What can however make this paper a stronger contribution is pursuing other sensor modalities (depth images, surface normals, segmentations, etc) and then exploring various appropriate losses there. As it stands, the paper is only applying a typical reconstruction-based loss to the proprio and contrastive to the image, which are the current norms in the field - no exciting surprise!
>
> The key insight and contribution of our work is that the effectiveness of additional information depends on the way that it is used. Both the “Concat” and “Joint” agents receive the same information, but the joint representation matches concatenation on simple tasks, and outperforms it as task complexity increases. Similarly, we show that the combination of commonly used individual reconstruction and contrastive losses is crucial for performance. This combination is novel, and we believe the insights that we present to be helpful for many of our peers. Our combined methods outperform concatenation in model-free RL and improve upon SOTA model-based RL methods for tasks with natural video backgrounds. Additionally, they can improve performance in novel tasks with complex observations, such as the occlusion and locomotion tasks, where single modality methods fail.
>
> We thank the reviewer for suggesting the use of other sensor modalities. We agree that experiments with different modalities would strengthen our contribution and currently run experiments using Depth Images and Proprioception for the OpenCabientDrawer Task. We will add these results to the final version of our manuscript.
>
> > Thirdly, even though this work provides extensive experiment results, in many figures, the methods showing the effect of each loss, for instance `Joint(R+R)` and `Joint(CV+R)`, the performance of these seem to be on-par with each other e.g. `Figure 8, Figure 2, Figure 3` etc.
>
> The methods only perform similar in the comparatively simple DeepMind Control Suite with standard observations (Figure 2 and Figure 8 in the original manuscript). While these results are important for a complete picture, we find the results on the harder tasks (Occlusins, Locomotion and OpenCabinetDrawer) to be more significant. We thank the reviewer for pointing out that this difference could be presented more clearly and put a clearer emphasis on more complex tasks in the revision. We additionally moved results for the DMC task with standard observations to the supplement.
>
> > Even though its nice the huge amount of analysis performed in this work, the concluding story is very hard to digest, specially since there are many different acronyms to their proposed method such as `Joint(R+R), Joint(CPC+R), Joint(CV+R)`. In addition, some baselines are missing for some tasks (e.g. `Figure 5`) and it makes drawing a final conclusion hard.
>
> We thank the reviewer for pointing this out. In the revision, we ensured that all acronyms were properly introduced in the “Representation Learning Methods” paragraph in Section 4 before presenting and discussing our results. We changed the reward curves to bar plots in the main part of the paper which allows a more comprehensive overview of all our results and added DrQv2, with and without proprioception, as a baseline for the OpenCabientDrawer Task (Figure 5). We believe that these changes make our findings and conclusion easier to digest, and would appreciate additional feedback if issues with them remain.
>
> > I also disagree with the following statement made in the paper `While many self-supervised representation ... neglect other available information, such as robot proprioception`. Adding proprioceptive state observations along with images is not novel in the field (especially in robotics).
>
> We agree that adding proprioceptive states to image observations is common in reinforcement learning with images, especially in robotics. However, existing approaches usually learn no explicit representations or concatenate an image representation with proprioception. Our work shows that using the proprioception to learn the representation in the first place, which is much less common, is usually beneficial. We clarified this distinction in the abstract.

---

> > ### Author Response · Authors · 2023-11-17
> >
> > > Except for combining multi-modal inputs with appropriate losses per each, are there any other interesting take-away observations from this work
> > >
> >
> > The main take aways of our work are as follows: (i) Joint representations outperform the concatenation of image representations and proprioception. (ii) Care has to be taken when choosing the individual losses and an appropriate combination can further improve performance. (iii) Combining standard losses provides an easy and effective approach to model-based RL if image reconstruction is not feasible.
> >
> > > Is there a reason many of the model-based baselines are missing for the `OpenCabinetDrawer` task?
> > >
> >
> > We omitted experiments with model-based approaches on the Locomotion Suite and OpenCabinetDrawer task because of the superior performance of their model-free counterparts in the previous experiments. We added DRQv2 with and without proprioception as a baseline for OpenCabinetDrawer in the revision.

---

### Official Review · Reviewer_ZjLm · 2023-11-01

**Soundness:** 4 excellent
**Presentation:** 4 excellent
**Contribution:** 3 good
**Rating:** 8
**Confidence:** 5

**Summary:**

The paper studies RL problems with multiple modalities of different nature, i.e., images and proprioception. The authors argue that reconstruction and contrastive objectives for representation learning, studied separately in prior work, are better tailored to each modality and combined in a joint fashion. This proposal is realized in the recurrent state-space model (RSSM) framework, where the authors extend the formulation to multiple modalities. To further highlight the strengths of each combination, the authors propose new datasets and tasks, and perform an extensive comparison against a variety of baseline models.

**Strengths:**

- Extends the RSSM model to multiple observation models to account for multi-modalities
- Introduces two datasets with specific additional challenges: VideoBackgrounds and Occlusions
- Introduces a new Locomotion benchmark with a focus on egocentric vision for obstacle avoidance (6 tasks)
- Performs additional experiments in the OpenCabinetDrawer task with variations in lighting and surroundings
- Extensive comparison over a large collection of model families

**Weaknesses:**

Nothing stands out, beyond the remaining questions pointed out in the limitations section, also raised in the questions below.

**Questions:**

**Missing technical discussion:**
- Is it possible to combine both contrastive learning paradigms? Or, to alternate between the two objectives at each epochs based on a given metric? Looking again at the discussion leading up to Eq.2 and Eq.3, there's no clear reason to favor one over the other. Moreover, as the two equations are pretty similar, perhaps it hints at a more general form. (Could the CV term be an alternative to the reward-based regularizer?). It's a useful ablation study to study CV or CPC in isolation, but since the experiments show distinct advantages to each formulation, it's likely the agent can learn to combine the two flavors. (Now I also see no reason why there's no Joint(CV + CPC) or Joint(CPC + CV) in the experiments. Makes me wonder why the authors completely overlook this option.)

**Presentation:**
- Abstract, last sentence: please clarify at this point what is meant by "common practice", as explained in the 3rd paragraph of the introduction.
- Section 3:
    - Suggest to break up the paragraph before Eq.2, possibly using a bold header corresponding to the block for CPC.
    - The inputs to the score functions seems to be flipped at the end of S3.1
- Section 4:
    - It would help to e.g. move the "Representation Learning Methods" paragraph to the beginning of the section before any of the figures to help read the legend.
    - ProrioSAC -> ProprioSAC

---

> ### Author Response · Authors · 2023-11-17
>
> We thank the reviewer for their time, effort, and especially for their general appreciation of our work. We incorporated their feedback regarding the presentation in the revised version of our paper and would like to use this opportunity to address their questions regarding a missing technical discussion.
>
> > Is it possible to combine both contrastive learning paradigms? Or, to alternate between the two objectives at each epochs based on a given metric? Looking again at the discussion leading up to Eq.2 and Eq.3, there's no clear reason to favor one over the other. Moreover, as the two equations are pretty similar, perhaps it hints at a more general form. (Could the CV term be an alternative to the reward-based regularizer?). It's a useful ablation study to study CV or CPC in isolation, but since the experiments show distinct advantages to each formulation, it's likely the agent can learn to combine the two flavors. (Now I also see no reason why there's no Joint(CV + CPC) or Joint(CPC + CV) in the experiments. Makes me wonder why the authors completely overlook this option.)
> >
>
> While an ad-hoc combination of the contrastive losses should be feasible in practice, we are not aware of any work or method that would theoretically justify such a combination. This work focuses on studying them in isolation, though we agree that the idea of a more general form that combines the benefits of both CV and CPC is very interesting and has potential for further improvements. In particular, a rigorous combination of both types of losses for a single modality seems promising.
>
> Using our naming system, Joint(CV + CPC) or Joint(CPC + CV) correspond to using a contrastive variational approach for the images while using a CPC approach for proprioception and vice versa.  We have not studied these possibilities as we believe that contrastive approaches for the low-dimensional, noise-free proprioception cannot improve over using reconstruction, i.e.  Joint(CV + R) or Joint(CPC + R).

---

> > ### Comment · Reviewer_ZjLm · 2023-11-21
> > **Acknowledgement**
> >
> > Thank you for the response and clarification.
> >
> > I may need to revise my scores after digesting the other rebuttals.

---

### Official Review · Reviewer_G7AG · 2023-11-03

**Soundness:** 3 good
**Presentation:** 3 good
**Contribution:** 2 fair
**Rating:** 5
**Confidence:** 3

**Summary:**

This paper studies the problem of reinforcement learning from observations collected across different sensors, and in this work, specifically, image observations and robot proprioception. The representation learning method aggregates historical observations through the Recurrent State Space Model (RSSM). The main difference from prior work is that the latent representation is now trained to be predictive of observations from multiple sensors. The experiments are conducted in various simulated vision-based environments, including an ego-centric cheetah-run task with obstacles.

**Strengths:**

- The experiments study a range of and introduce some new environments, from locomotion to manipulation with moving backgrounds and with occlusions. These settings are particularly challenging because they rely (1) on robust representation learning and (2) on both vision and proprioception.
- The proposed approach tackles both of these challenges by learning a representation that is task-relevant and represents both modes of observation.
- The observation that the joint representation leads to more efficient RL over concatenation is useful for practitioners.

**Weaknesses:**

- The experiments only look at image observations and proprioception as the two modalities. It would be interesting to see this approach applied to other sensor modalities.
- The extension of RSSMs to model both image observations and proprioception is a straightforward one, which is the primary contribution of this work.
- It seems like the correct loss for each modality varies quite a bit across domains.
- I'm still unclear on details for some of the comparisons and results (see Questions).

**Questions:**

- Why do you think there is such a gap between Joint and Concat, where Concat performs the same as ProprioSAC on the cabinet tasks? It seems like Concat should be able to produce any representation that Joint can. Does Concat eventually converge to the same performance as Joint in Fig. 5 if we let it train for longer?
- Do the DenoisedMDP and DreamerPro comparisons utilize observations from both modalities?
- In Fig. 6 (right), are the images reconstructed from a separately trained decoder as a way to probe the representations?
- How are the losses for different observation modalities weighted against each other?

---

> ### Author Response · Authors · 2023-11-17
>
> We thank the reviewer for their time and effort and appreciate their recognition of our work in addressing complex issues and offering valuable insights for practitioners in the field. The revised version addresses several of their concerns, yet we would like to further address them in the following:
>
> >[...] It would be interesting to see this approach applied to other sensor modalities.
>
> We agree that experiments on more modalities would strengthen the message of our paper and are currently experimenting with the combination of depth images and proprioception on the OpenCabinetDrawer task. We will add the results to the final version of our manuscript upon acceptance. In addition, our “proprioception” modality already includes different sensors such as joint encoders, data from internal measurement units (IMUs, i.e. global velocities), and forces.
>
> > The extension of RSSMs to model both image observations and proprioception is a straightforward one, which is the primary contribution of this work.
>
> Extending the RSSM to use multiple modalities is only a small part of our contribution. We think providing a unified framework to combine different loss functions for different modalities is more important and such a combination has not been considered before. Our experiments demonstrate that this combination is the main factor behind the benefits of our approach. This finding is particularly relevant given that practitioners typically employ concatenation or single modality methods, presumably because simply adding proprioception without careful consideration often fails to yield significant benefits. We thus agree with the reviewer that these insights are beneficial for practitioners in the field.
>
> > It seems like the correct loss for each modality varies quite a bit across domains.
>
> We believe that the differences between reconstruction-based and contrastive image losses over different tasks and domains are expected and well explainable by intuition and theory: As reconstruction enforces the model to capture all information in the observation, it yields good representations if all information is relevant. If that is not the case, reconstruction focuses too much on irrelevant aspects that can be ignored by contrastive approaches.  We note that exploring the optimal choice of contrastive loss, though intriguing, is not the primary goal of our current research. That said, we believe that this is an interesting direction for future research for example by combining both paradigms, as also suggested by Reviewer ZjLm.
>
> > Why do you think there is such a gap between Joint and Concat, [...]
>
> While both Joint and Concat receive the same information and thus should indeed be able to produce the same representations, we believe that including the proprioception in the representation learning guides the representation learning toward more relevant aspects of the image. The qualitative results in Figure 6 aim to provide some evidence and intuition for this. These results indicate that including the proprioception in representation learning helps the model to both ignore irrelevant aspects of the image and better capture the relevant aspects. We were unfortunately so far unable to run the OpenCabinetDrawer experiments for longer and thus cannot decisively answer the question regarding Fig 5.  Even if the concat baseline would eventually catch up, there are still large sample complexity benefits. Furthermore, we saw in several other experiments (Occlusions and Locomotion) how the concat baselines converge to policies that achieve less reward than the Joint Representation approach.
>
> > Do the DenoisedMDP and DreamerPro comparisons utilize observations from both modalities?
>
> Except for DrQv2 (I+P), the baselines use image modalities, which they were originally designed for. We use the code bases provided by the respective authors, which do not allow the use of multiple modalities. The baselines are meant to establish the difficulty of the new tasks and show that using joint representations can provide SOTA results. While the baselines can likely be improved with joint representations, such an improvement would further highlight the benefits of joint representation and thus would be in the spirit of our main message.
>
> > In Fig. 6 (right), are the images reconstructed from a separately trained decoder as a way to probe the representations?
>
> Yes. The decoder is trained entirely post-hoc to probe the representations and generate the plots. No information flows back from it to the original model. We clarified this in the revision.
>
> > How are the losses for different observation modalities weighted against each other?
>
> Except for the weighted KL for the CPC-Objective in Equation 4, there are no explicit weighting factors in the objectives. For the OpenCabinetDrawer task, we implicitly weigh the proprioception reconstruction by assuming a lower variance for the respective decoder. See Appendix B.2., for details.

---

### Official Review · Reviewer_nyC5 · 2023-11-04

**Soundness:** 3 good
**Presentation:** 2 fair
**Contribution:** 2 fair
**Rating:** 3
**Confidence:** 4

**Summary:**

This paper proposes a framework to jointly learn representations from vision and proprioception sensors based on the recurrent state space model (RSSM). It systematically studies the ways to combine contrastive and reconstruction losses on different sensor inputs through comprehensive experiments.

**Strengths:**

1. In general, the writing is clear and easy to follow
2. The experiments are solid and comprehensive. The experiment section and appendix present the results of various joint representation learning designs (CV, CPC, reconstruction) as well as different ablations (concat, image-only, state-only) and baselines (model-free, model-based) under different environment settings.
3. Code is provided with good reproducibility.

**Weaknesses:**

1. While the overall logic is clear and smooth, some specific notations and figures are confusing.

    (a) From the figure plotted in the main paper or appendix, it's very hard to draw any conclusions about which representation learning method is the best. Instead of presenting the curves, drawing some bar charts about the final performance average over different settings and environments can be more straightforward.

    (b) The figures in the main paper (Figure 2, 3, 4) are interleaving with model-free/model-based and occlusion/locomotion, which makes it hard to understand what's been delivered

    (c) There is no explanation of e.g, Joint(CV+R), Joint(CPC+R) which make the confusion that the "+R" is for reward reconstruction.

2. The results are not clear or convincing enough to draw a strong conclusion as in the discussion section

    (a) The environmental and experimental design is not delivered clearly. Please address Questions 1. a for clarification.

    (b) "In the more difficult settings, i.e., Occlusions, Locomotion (Fig. 4), and OpenCabinetDrawer (Fig. 5), using a joint representation gives the largest benefits", "In the Locomotion experiments, the CPC approaches (Fig. 4) have a significant edge over reconstruction", which is only true for model-based occlusion (Fig. 4), and for Fig. 5 the gain is unclear (see question 1. b).

    (c)  "In the Locomotion experiments, the CPC approaches (Fig. 4) have a significant edge over reconstruction". First, the gain is not significant. Second, Locomotion's model-based results are missing.

    Understandably, there may not exist a unified framework that works best for model-based and model-free RL. Given that many details are missing, the conclusion seems too strong and not rigorous enough. Some possible improvements e.g. separately discuss (i) model-free and model-based, (2) locomotion and manipulation, and (3) standard image and background changes, to make a less strong but more rigorous conclusion. Also DMC's results are very saturated, it might be more convincing to include more diverse domains (e.g. more Maniskill/RLBench/FrankaKitchen results).

3. The formulation in equation (4) is not mathematically rigorous. If CPC is applied to estimate the MI between the current representation and the next observation, the KL part should be factorized differently but not naively apply equation (1).

4. Typo in caption: Figure 9 should be model-based results.

**Questions:**

1. Task and experimental details

    (a) Is the "standard image" /"video background" / "occlusion" suite all modified from environments in Table 1? Are the curves averaged in each suite? How many seeds do you run for each task? The curves in each figure have low variance, which doesn't look like an average as different tasks require very different sample complexities in each suite. If you normalized that, how the normalization was done?

    (b) Are the Maniskill results model-free or model-based? Why only a SAC baseline is included?

    (c) Why the locomotion's model-free results are missing?

2. Model details

    (a) For the model-free results, are you also reconstructing reward based on latent representation z?

---

> ### Author Response · Authors · 2023-11-17
>
> We thank the reviewer for their time and effort, and particularly for acknowledging our “solid and comprehensive” experiments. We appreciate them pointing out the typos and ambiguities in our initial submission and address them in our revision. We acknowledge the presentation issues (c.f. Weaknesses 1.) in the original manuscript. Following the reviewer’s suggestion, we introduce bar charts for all experiments in the revision and avoid discussing multiple task suits in a single figure. To summarize our experiments, we conduct model-free and model-based experiments for the standard DMC tasks as well as those with modified images (video background and occlusion). We focus on model-free approaches for the Locomotion Suite and OpenCabinetDrawer as they showed better performance than their model-based counterparts in the DMC tasks with all types of images. We included DRQ-v2  as a baseline for OpenCabinetDrawer in the revision to also provide the compressions to SOTA related work for this task.
>
> > The environmental and experimental design is not delivered clearly. Please address Questions 1. a for clarification. (a) Is the "standard image" /"video background" / "occlusion" suite all modified from environments in Table 1?
>
> All three suites contain the 7 environments listed in Table 1 and differ only in the image observations. We clarified this in the revision.
>
> > Are the curves averaged in each suite? How many seeds do you run for each task? The curves in each figure have low variance, which doesn't look like an average as different tasks require very different sample complexities in each suite. If you normalized that, how the normalization was done?
>
> The results are aggregated over each suite. We ran 5 seeds per task for the locomotion and DMC tasks, so each curve/bar represents a total of 30 runs for the locomotion suit and 35 runs for the other three DMC suits. We use interquartile means and stratified bootstrapped confidence intervals for the error bars. This procedure follows the widely adopted (in RL) suggestions from Argawal et al (2021) and is specifically designed to aggregate results over an experiment suite in a principled manner. We compute the results using the code published by Argawal et al and provide per-task results for completeness in Appendix C. For OpenCabientDrawer we use 20 seeds to establish significance.
>
> > Also DMC's results are very saturated, it might be more convincing to include more diverse domains (e.g. more Maniskill/RLBench/FrankaKitchen results).
>
> The occlusion and locomotion tasks are novel tasks building on DMC that we designed to study hard information fusion problems in representation learning for RL. While natural video backgrounds have been previously used for a similar purpose, both related work and our experimental results indicate that there is still room for improvement. We are currently running additional Maniskill experiments, investigating the combination of Depth Images and proprioception. We will add those to the final version of our paper.
>
> > (b) "In the more difficult settings, i.e., Occlusions, Locomotion (Fig. 4), and OpenCabinetDrawer (Fig. 5), using a joint representation gives the largest benefits"
> (c) "In the Locomotion experiments, the CPC approaches (Fig. 4) have a significant edge over reconstruction".
>
> > Understandably, there may not exist a unified framework that works best for model-based and model-free RL. Given that many details are missing, the conclusion seems too strong and not rigorous enough. Some possible improvements e.g. separately discuss (i) model-free and model-based, (2) locomotion and manipulation, and (3) standard image and background changes, to make a less strong but more rigorous conclusion.
>
> We apologize for the confusing presentation in our original submission. The updated bar plots of the revision show the benefit of the joint representation for all mentioned cases. The new bar plots also clarify that the CPC approaches have an edge over the reconstruction approaches on the locomotion tasks. For example, Joint(CPC+R) achieves a reward of almost 800 while Joint(R+R) achieves only slightly more than 600. Further, we made our statements more precise and updated the references to the new figures in the revision. We ensured all our claims are backed up by experimental findings and added missing details for the experimental description (number of seeds, aggregation) and baselines (DrQv2 for ReplicaDrawer).
>
> > The formulation in equation (4) is not mathematically rigorous. [...]
>
> Adding the weighted KL term to the CPC objective is based on theoretical (Shu et al, 2021)) and empirical (Nguyen et al (2021),  Srivastava et al (2021)) findings which highlight the importance of enforcing consistency between prior and posterior beliefs when training state space representations with predictive losses.  We conducted preliminary experiments further validating these findings and decided to adopt the method of Srivastava et al (2021).

---

> > ### Author Response · Authors · 2023-11-17
> >
> > > (c) Why the locomotion's model-free results are missing?
> >
> > We ran only model-free experiments for locomotion, as model-free methods outperformed model-based ones on the DMC. We thank the reviewer for pointing out the mistake in the caption of Figure 12 and fixed it in the revision.
> >
> > > Model details: (a) For the model-free results, are you also reconstructing reward based on latent representation z?
> >
> > Yes. For all model-free experiments, we follow the findings of Tomar et al (2023), which show reward reconstruction for representation learning helps, even if the policy learning can work ground truth rewards.
> >
> > **References**
> >
> >
> > Agarwal et al. Deep reinforcement learning at the edge of the statistical precipice, 2021
> >
> > Shu et al . Predictive coding for locally-linear control, 2020
> >
> > Nguyen et al. Temporal predictive coding for model-based planning in latent space, 2021
> >
> > Srivastava et al. Robust robotic control from pixels using contrastive recurrent state-space models, 2021
> >
> > Tomar et al.  Learning representations for pixel-based control: What matters and why?, 2023

---

### Author Response · Authors · 2023-11-17

We thank all reviewers for their time and constructive feedback. We uploaded a revision of our work based on the reviewers' feedback and will also address their reviews individually.

The most significant change in the revised version concerns the presentation of our results. Based on the suggestions of reviewer nyC5 we replaced the loss curves in the main part of the paper with bar charts, showing the aggregated final performance of all approaches. This allows us to present a more comprehensive overview of all our experiments and enhances clarity. The reward curves and performance profiles for all experiments are still available in Appendix C. Further, we marked major changes in text in blue, so they are easily identifiable. These changes include:

- Added missing DrQ-v2 Baselines for OpenCabinetDrawer
- Clarification of statements in the abstract
- Clarified motivation for reward reconstruction in model-free RL, KL term in CPC Objective, and its weighting.
- Reordering of paragraphs in sections 3 and 4. All acronyms are now introduced before presenting or discussing results.
- Added details about the number of seeds and aggregation of results.
- Fixed typos. We thank the reviewers for pointing those out.

Additionally, we are currently running experiments with the OpenCabinetDrawer Tasks and Depth Images to further strengthen our message by including different modalities. We plan on adding those results to the final version of the manuscript.

We hope we could address the reviewers’ concerns in our revision and individual answers and invite them to point out remaining ambiguities or unclear aspects.

---

### Author Response · Authors · 2023-11-20
**Added Depth Image Experiments**

Dear reviewers and AC,

we have now uploaded another revision of our paper, including new experiments with depth images on the OpenCabinetDrawer Task.
We again thank the reviewers VgNz and G7AG for suggesting such experiments. As detailed in the revision, they further showcase the benefits of joint representations, and we believe they strengthen the message of our work.

---

### Author Response · Authors · 2023-11-22

We thank the reviewers again for their valuable feedback and suggestions. We also thank reviewer ZjLm for replying to our comments. Given the approaching end of the discussion phase, we hope we could address the reviewers' concerns appropriately and kindly ask them for further feedback. We believe, such feedback would allow a better assessment, and if needed further improvements, of our work. Thank you again for your time, effort, and consideration.

---

### Comment · Area_Chair_bo98 · 2023-11-23
**Author-Reviewer discussion period ending *very* soon**

Thank you to the reviewer ZjLm for responding. The authors have put great effort into their response, so can I please urge **all other reviewers to answer the rebuttal**.
Thank you!

---

### Meta-Review · Area_Chair_bo98 · 2023-12-05

**Metareview:**

This research tackles the challenge of combining various sensor inputs in reinforcement learning. By leveraging robot proprioception alongside image-based learning, they introduce a novel approach that combines reconstruction-based and contrastive losses tailored to each sensor modality.

Reviewer nyC5 noted concerns about clarity in figures and notations, expressing difficulties in interpreting certain data and inconsistencies between stated conclusions and actual observations. Additionally, there were concerns about incomplete experimental design, with insufficient contextual information impacting the robustness of the drawn conclusions. Reviewer G7AG emphasised the limited exploration of sensor modalities, specifically pointing out the paper's confinement to image and proprioception without offering substantial innovation by considering other sensor types. This was addressed in the rebuttal by adding in depth information as an additional modality. The perceived lack of novelty in the primary contributions was highlighted, suggesting a straightforwardness in approach without significant inventive elements. Furthermore, inconsistencies in determining appropriate loss functions for different modalities across domains raised doubts about the methodology's consistency and adaptability. Reviewer VgNz echoed concerns about predictability in the results, where observed improvements using joint representations lacked surprise, aligning with expectations. The innovative contribution regarding loss functions for modalities was seen as weak, as it seemed to adhere closely to existing norms without breaking new ground. The comprehension challenges stemming from excessive acronyms and missing baseline comparisons made it difficult to fully grasp the paper's conclusion.

Of the four reviews, only one supported the paper to be accepted, but this was not championed during the rebuttal phase. I have read and acknowledged the authors message to the AC. The message weighed heavily on the fact that despite the joint representations been seen as “straightforward” (G7AG) and “not surprising” (VgNz), a large-scle study is still valuable.This AC agrees, but shares similar views to nyC5, that DMC's results are very saturated, and that it might be more convincing to include more diverse domains (e.g. more Maniskill/RLBench/FrankaKitchen/etc results).

**Justification For Why Not Higher Score:**

Given that the novelty is on the slim side, myself and reviewers expect to see strong and through experimental analysis, which this paper was lacking.

**Justification For Why Not Lower Score:**

N/A

---

### Decision · Program_Chairs · 2024-01-16

Reject